# LACE: Lightweight Attribution-guided Concept Evolution for Continual Learning

## Abstract

We address *concept proliferation* in interpretable continual learning and present **LACE**, a lightweight framework that couples Concept Bottleneck Models with a learnable concept-alignment layer, *Concept Attribution* (CA) that quantifies per-concept importance under standard attribution axioms, and *Concept Verification* (CV) that selects pruning budgets via a data-reusable approximation to leave-one-out with an IRLS hat-matrix correction. A prototype-augmentation mechanism stabilizes learning without exemplars. Across coarse- and fine-grained benchmarks, LACE yields compact, reusable concept sets, consistently improves or matches strong baselines, and narrows the gap between average and last-task accuracy, offering an auditable and parameter-efficient route to continual concept evolution. Our code is available at: https://anonymous.4open.science/r/LACE-7FD6/.

## 1 Introduction

Continual Learning (CL) aims to acquire new knowledge from a sequence of tasks or stages (Hadsell et al., 2020) while mitigating catastrophic forgetting (Kirkpatrick et al., 2017) and preserving stable reasoning. CL is crucial for open-world deployment, privacy-constrained settings, and cost-sensitive training. However, *most* CL methods are still evaluated primarily by external behavioral metrics (e.g., accuracy or forgetting), with limited *auditable characterization of how the model's decision basis evolves across tasks*. This gap weakens interpretability and impedes a systematic analysis of "why forgetting is avoided" and "where knowledge is retained."

To address this, *Concept Bottleneck Models* (CBMs) (Koh et al., 2020) have been introduced into the CL context: an explicit, human-interpretable concept layer is inserted between features and the classifier so that the model first predicts concepts and then makes the final decision, thereby providing auditable and intervenable evidence along the reasoning chain. Building on this idea, language- and multimodal-guided variants further leverage large pretrained models (e.g., CLIP (Radford et al., 2021) and large language models, LLMs (Brown et al., 2020)) to align vision–language semantics, using "concept presence" both for explanation and classification as cross-task *semantic anchors* (Yuksekgonul et al., 2023). Integrating CBMs into CL (henceforth CBM-CL) (Yu et al., 2025) brings two immediate benefits: (i) historical concepts can be reused while new concepts are incrementally discovered, which helps alleviate catastrophic forgetting; and (ii) the temporal evolution of the decision basis becomes traceable—one can continuously monitor which old concepts are preserved and which new concepts are introduced—thus improving interpretability and auditability.

Yet existing work often suffers from *concept proliferation*: from a human perspective, classic cognitive psychology suggests a working-memory capacity of roughly $7\pm2$ units (Miller, 1956) (not a strict cap, but indicative of a preferred evidence scale per decision). Too many concepts increase cognitive load, introduce redundancy or spurious correlations, and reduce interpretability and usability. From a model perspective, redundant concepts dilute representations, amplify noise accumulation and overfitting risks, and obscure key rules, thereby weakening generalization. From a CL pipeline perspective, if each incoming task introduces many unconstrained concepts, they tend to compete with existing ones and accelerate forgetting (e.g., new-task concepts may reuse the same feature channels and overwrite weights supporting old concepts, causing the model to "forget" prior knowledge). Therefore, *controlling concept cardinality without sacrificing discriminative power* is pivotal to making CBM-CL practical. We propose a concept-centric CBM-CL framework (flowchart as shown in Figure 1), **LACE** (Lightweight Attribution-guided Concept Evolution), and

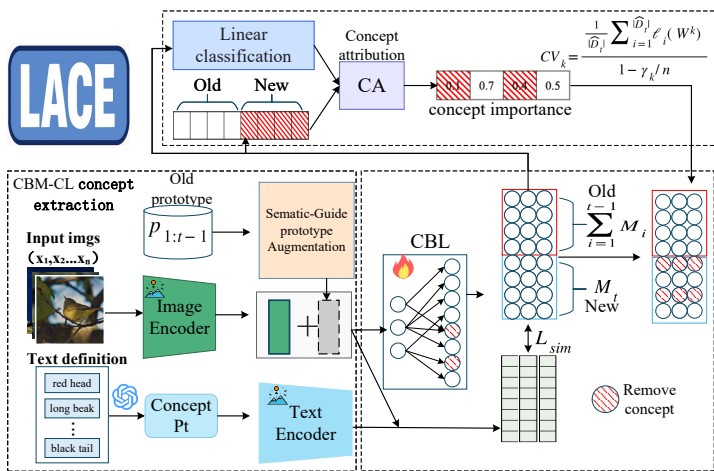

Figure 1: Structure of Lightweight Attribution-guided Concept Evolution for Continual Learning (LACE)

instantiate a closed-loop *concept pruning* process operating in three phases—*attribution*, *selection*, and *verification*. Concretely, at the start of each incremental task, we expand the current concept pool using language and multimodal priors, and align image representations to the concept space through the bottleneck so that "concept scores" are both human-interpretable and directly drive classification. We then perform *Concept Attribution* (CA) (Wu et al., 2020): we assess each concept's marginal impact on the prediction at the concept layer, adhering to basic principles of consistency and stability to obtain a reproducible, comparable "importance scale." Based on this scale, we conduct *concept selection and pruning*: we preferentially retain cross-task reusable key concepts while removing redundant or spurious ones, compressing to a size closer to human-readability. To avoid over- or under-pruning, we further perform *Concept Verification* (CV) (Mangal et al., 2024): without relying on external data, we compare generalization across pruning scales and automatically determine the number of concepts to retain, yielding a clear trade-off among performance, interpretability, and stability. In parallel, we introduce a *prototype augmentation* mechanism to suppress forgetting: under exemplar-free constraints, we synthesize lightweight pseudo-feature replay using prior-task class semantics and current-task representative features as stable anchors that regularize updates, preventing destructive overwriting of old knowledge. Overall, LACE uses *concepts* as a unifying interface, embedding human-readable semantic anchors into both training and selection: for humans, controlled concept size reduces reading and auditing burden; for models, de-redundancy and stable anchors improve generalization and temper noise accumulation; for the pipeline, CA and CV are executed within the existing training data, with controlled overhead and no extra annotation.

**Contributions.** (i) We systematically identify the threefold impact of *concept proliferation* in interpretable CL—diminished human readability, diluted effective representations, and heightened forgetting risk—and propose a *concept pruning* framework. It employs *Concept Attribution* (CA) to quantify a concept's marginal contribution to predictions and uses *semantic anchors* to integrate CBMs into CL (CBM-CL), removing redundant concepts and retaining a compact, high-utility set *without* additional supervision. (ii) We provide a theoretical characterization and a practical rule for selecting the *pruning scale* via *Concept Verification* (CV), determining how many concepts to keep under a fixed data budget. This yields a transparent trade-off among performance, interpretability, and stability, and offers auditable selection criteria for semantic trimming in CBM-CL. (iii) Across multiple CL benchmarks, we demonstrate the feasibility and generality of the framework: it maintains or improves task accuracy while substantially reducing concept count, lowering forgetting metrics, and enhancing explanation readability and cross-task consistency. We will release full code and configurations to facilitate reproduction and extension.

## 2 RELATED WORK

### 2.1 CONTINUAL LEARNING

Continual Learning (CL) (Hadsell et al., 2020) seeks to acquire new knowledge over a sequence of tasks while minimizing degradation of previously learned capabilities. Classical approaches fall into three families: *regularization-based* methods constrain drift via knowledge distillation or parameter-importance penalties (e.g., LwF, EWC) (Li & Hoiem, 2016; Kirkpatrick et al., 2017); *replay-based* methods consolidate prior representations through sample or feature replay (e.g., iCaRL) (Rebuffi et al., 2017); and *architectural* methods reduce cross-task interference via parameter isolation or network expansion (e.g., Progressive Neural Networks) (Rusu et al., 2016). With the rise of large-scale pretrained models and vision Transformers, *prompt-based/parameter-efficient* CL has gained traction: L2P, DualPrompt, and CODA-Prompt achieve strong performance across incremental settings without storing old samples by introducing only a small number of prompt parameters (Wang et al., 2022a;b; Smith et al., 2023). Nevertheless, most methods are still assessed primarily by *external behavioral metrics* (e.g., accuracy or forgetting), with limited *auditable characterization of how the model's decision basis evolves over tasks*. Motivated by this gap, recent work integrates *interpretable structure* into CL pipelines. For instance, ICICLE leverages prototypical parts and interpretability regularization to balance accuracy and interpretability in class-incremental learning (Rymarczyk et al., 2023), suggesting that *semantic-layer constraints* can be effective levers for mitigating catastrophic forgetting.

### 2.2 CONCEPT BOTTLENECK MODELS (CBMs), CONCEPT CARDINALITY, AND PRUNING

Concept Bottleneck Models (CBMs) insert an explicit *concept layer* between features and the classifier, predicting *human-interpretable concepts* prior to the final decision and thereby providing intervenable, auditable evidence along the reasoning chain (Koh et al., 2020). When concept labels are unavailable or a black-box backbone is preferred, *post-hoc* PCBM transfers a concept space across datasets or via *language–vision* alignment, retaining CBM's interaction advantages (Yuksekgonul et al., 2023). Pushing this further, *language/multimodal-guided* CBMs use LLMs to propose concept candidates and CLIP to align text–image semantics, letting concept scores serve both explanation and classification, with promising generalization and stability in practice (Yang et al., 2023). Building on this line, *language-guided CBMs for CL* have been systematically explored, maintaining concept–class semantic consistency during incremental training to jointly improve interpretability and resistance to forgetting (Yu et al., 2025).

Controlling the *cardinality* of the concept layer is critical for readability, robustness, and computational cost: too few concepts limit coverage and discriminability, whereas too many induce redundancy and spurious correlations that degrade explanation quality. The literature highlights the need for *parsimonious retention* from multiple angles. For example, prototype-based interpretable networks (ProtoPNet, ProtoTree) express "this looks like that" via sparse prototypes/paths at training or inference time (Chen et al., 2019; Nauta et al., 2021). In CL settings, the CBM-CL paradigm is thus natural: use the *concept layer* as a cross-task semantic anchor and pair it with *pruning* or sparse selection to suppress redundancy so that explanation quality and predictive performance reinforce each other.

### 2.3 ATTRIBUTION AND CONCEPT ATTRIBUTION (CA)

Attribution methods quantify the contribution of input evidence to predictions. Main paradigms include gradient/path-based *Integrated Gradients* (IG), which formalizes consistency via *Sensitivity* and *Implementation Invariance*; feature-map–based *Grad-CAM* for class-discriminative localization; and black-box estimation via randomized masking, as in *RISE* (Sundararajan et al., 2017; Selvaraju et al., 2017; Petsiuk et al., 2018). Building on these, *MFABA* introduces a decision-boundary view for faithful and efficient explanations that respect attribution axioms while substantially accelerating computation, often improving or maintaining explanation quality with order-of-magnitude speedups (Zhu et al., 2024).

Inspired by this line of work, we *lift pointwise attribution to the concept space* and propose **Concept Attribution (CA)**: treating concepts as atomic units and estimating their marginal impact on the model output. Concretely, we leverage MFABA's boundary insights and efficiency to reduce the cost

of concept-layer ablations/perturbations while remaining consistent with attribution axioms. The goal of CA is not another visualization per se, but a stable, auditable *importance scale* that supports downstream *concept selection*, *stability analysis*, and *cross-task sharing*.

## 2.4 SELECTING THE PRUNING BUDGET (CV)

Interpretable learning with concepts calls for a *data-driven* determination of the post-pruning concept *quantity*. We introduce **Concept Verification (CV)**, inspired by *leave-one-out cross-validation (LOOCV)*: under *no external data* assumptions, CV compares generalization across pruning scales via leave-one evaluation, selecting the number of concepts to retain (Stone, 1974; Arlot & Celisse, 2010). As in classical cross-validation, CV provides relatively unbiased risk estimation under limited data budgets; combined with the stable concept ranking induced by CA, it enables a structured search over a small candidate set to obtain a principled trade-off among accuracy, interpretability, and computational cost. In summary, the $CA \rightarrow CV$ pipeline decouples yet couples "importance estimation" and "cardinality selection": CA ensures *faithful, auditable* grounds, while CV ensures *robust, reusable* pruning decisions—together serving the goals of CBM-CL.

## 3 METHOD

### 3.1 PROBLEM DEFINITION

In the class-incremental learning (CIL) setting, a model is required to learn new classes sequentially over a series of tasks while retaining knowledge from earlier tasks (Yu et al., 2025). Let the task index set be $\mathcal{T} = \{1, 2, \ldots, T\}$. For task $t$, the dataset is

$$\mathcal{D}_t = \{(x_i, y_i)\}_{i=1}^{N_t}, \quad x_i \in \mathcal{X}_t, \ y_i \in \mathcal{Y}_t, \tag{1}$$

where $N_t$ denotes the number of samples in task $t$ and $\mathcal{Y}_t$ is the label set for task $t$. The label sets across different tasks are disjoint: $\mathcal{Y}_i \cap \mathcal{Y}_j = \varnothing \quad (\forall\, i \neq j)$. After completing $t$ tasks, the cumulative label space is $\mathcal{Y}_{1:t} = \mathcal{Y}_1 \cup \mathcal{Y}_2 \cup \cdots \cup \mathcal{Y}_t$, with the total number of classes given by $|\mathcal{Y}_{1:T}| = C$. During training on task $t$, the learner has access only to $\mathcal{D}_t$, whereas at inference time it must predict over $\mathcal{Y}_{1:t}$.

For representation learning, we adopt the CLIP framework (Radford et al., 2021) and introduce an image encoder $E_I(\cdot)$ and a text encoder $E_T(\cdot)$. Both map images and texts, respectively, into a shared semantic space with output dimensionality $d$.

### 3.2 ARCHITECTURE OF CBM-CL

This subsection presents the CBM-CL pipeline, which proceeds in three stages: **concept selection**, **concept alignment**, and **prototype augmentation**.

**Concept generation and selection.** For task $t$, we employ an LLM to propose semantic concepts for each class in $\mathcal{Y}_t$ (Oikarinen et al., 2023). This yields a task-specific concept pool $\mathcal{P}_t = \{c_1^t, c_2^t, \ldots, c_{M_t}^t\}$, where $M_t$ is the number of concepts for task $t$. As the sequence progresses, the cumulative concept pool is $\hat{\mathcal{P}}_t = \mathcal{P}_1 \cup \mathcal{P}_2 \cup \cdots \cup \mathcal{P}_t$.

To leverage these concepts, we encode them with CLIP's text encoder $E_T(\cdot)$ to obtain a bank of concept vectors $\mathbf{C}_t = \{E_T(c_1^t), E_T(c_2^t), \ldots, E_T(c_{M_t}^t)\}, \quad \mathbf{C}_t \in \mathbb{R}^{M_t \times d}$, with embedding dimension $d = 512$. The cumulative concept-vector bank is $\hat{\mathbf{C}}_t = \mathbf{C}_1 \cup \mathbf{C}_2 \cup \cdots \cup \mathbf{C}_t, \qquad \hat{\mathbf{C}}_t \in \mathbb{R}^{\left(\sum_{i=1}^t M_i\right) \times d}$.

**Concept alignment.** CBM-CL trains a Concept Bottleneck Layer (CBL) to map image features into the corresponding concept space. The CBL is a learnable linear transform with parameters $\mathbf{W}_c$. Because the concept space expands with each new task, the parameter size grows accordingly. At task $t$, $\mathbf{W}_c^t \in \mathbb{R}^{\left(\sum_{i=1}^t M_i\right) \times d}$, $d = 512$. Upon entering task $t{+}1$, we expand the parameters as $\mathbf{W}_c^{t+1} = \begin{bmatrix} \mathbf{W}_c^t \\ w^{\text{init}} \end{bmatrix}, \qquad \mathbf{W}_c^{t+1} \in \mathbb{R}^{\left(\sum_{i=1}^{t+1} M_i\right) \times d}$, where $w^{\text{init}}$ initializes the rows for the newly added concepts.

We optimize $(\mathbf{W}_c^t)^\top$ with the similarity loss

$$\mathcal{L}_{\text{sim}} = \cos\left(\left(E_I(x_i)\,(\mathbf{W}_c^t)^\top\right)^3,\ \left(E_T(x_i)\,\hat{\mathbf{C}}_t\right)^3\right), \tag{2}$$

which aligns the learned concept scores with CLIP's concept space. Using raw scores often yields many mid-range similarities (0.3–0.6), producing insufficient sparsity and salience; the cubic sharpening emphasizes high scores and suppresses low scores to obtain a peakier distribution (Yu et al., 2025).

We further introduce a linear head from concepts to classes with parameters

$$\mathbf{W}_l^t \in \mathbb{R}^{\left(\sum_{i=1}^t M_i\right) \times |\mathcal{Y}_t|}, \qquad \mathbf{W}_l^{t+1} = \begin{bmatrix} \mathbf{W}_l^t \\ w^{\text{init}} \end{bmatrix} \in \mathbb{R}^{\left(\sum_{i=1}^{t+1} M_i\right) \times |\mathcal{Y}_{t+1}|}. \tag{3}$$

For a sample $x_i$, the prediction is $\hat{y}_i = E_I(x_i)\,(\mathbf{W}_c^t)^\top\,\mathbf{W}_l^t$.

We train with cross-entropy $\mathcal{L}_{\text{ce}}(\hat{y}_i, y_i)$, and the overall objective is $\mathcal{L} = \mathcal{L}_{\text{ce}}(\hat{y}, y) + \lambda\,\mathcal{L}_{\text{sim}} + \sigma\,\mathcal{L}_{\text{sparse}}$, where $\lambda$ and $\sigma$ balance the terms, and $\mathcal{L}_{\text{sparse}}$ regularizes the concept activations to reduce overfitting (Yu et al., 2025).

**Prototype augmentation.** To mitigate catastrophic forgetting, we augment training with class prototypes. For each class in the current task,

$$P_i = \frac{1}{|\mathcal{D}_t^i|} \sum_{x \in \mathcal{D}_t^i} E_I(x), \qquad i \in \mathcal{Y}_t, \tag{4}$$

where $\mathcal{D}_t^i$ is the subset of $\mathcal{D}_t$ with label $i$. Because prototypes are defined in CLIP's image space, we match historical classes using their text embeddings. For a past class $j \in \mathcal{Y}_{1:(t-1)}$, we find its closest class index in the current task: $h_j = \arg\max_{i \in \mathcal{Y}_t} \langle E_T(y_j),\ P_i \rangle$.

Using images from class $h_j$, we construct pseudo-features for the old class $j$. Let $\mathbf{V}^{h_j} = \{E_I(x) \mid x \in \mathcal{D}_t^{h_j}\}$, then the pseudo-features are

$$\mathbf{V}_{\text{pseudo}}^j = \underbrace{P_j}_{\text{old class information}} + \underbrace{\left(\mathbf{V}^{h_j} - P_{h_j}\right)}_{\text{new class adjustment}}, \tag{5}$$

where $P_j$ is the prototype of old class $j$, $P_{h_j}$ is the prototype of the matched current class $h_j$, and $\mathbf{V}^{h_j}$ denotes the set of feature vectors from current class $h_j$. This equation defines a set of pseudo-features for old class $j$: each feature in $\mathbf{V}^{h_j}$ is translated from its corresponding prototype $P_{h_j}$ to the old class prototype $P_j$. Since all elements in $\mathbf{V}_{\text{pseudo}}^j$ reside in the same feature space as $E_I(x)$, they can naturally serve as the input feature set when constructing augmented training data in Equation (5). Note that $\mathbf{V}_{\text{pseudo}}^j$ represents a collection of feature vectors rather than a single prototype.

We mix these pseudo-features with the current data to define an augmented set

$$\mathcal{D}_t^a = \{(x_i, y_j) \mid x_i \in \mathbf{V}_{\text{pseudo}}^j,\ j \in \mathcal{Y}_{1:(t-1)}\}, \tag{6}$$

and the final training set is

$$\hat{\mathcal{D}}_t = \mathcal{D}_t \cup \mathcal{D}_t^a. \tag{7}$$

In practice, feature-mixed data are combined directly with $\{E_I(x_i) \mid x_i \in \mathcal{D}_t\}$ since both lie in CLIP's image space; the above exposition abstracts this for notational clarity.

**Intuition.** Before detailing the estimation and pruning logic, we outline the core intuition behind LACE. Concept proliferation in CIL creates a dilemma: retaining all concepts leads to redundancy and interference, while heuristic pruning risks losing critical information. LACE resolves this via a two-stage *Rank-and-Verify* process. First, **Concept Attribution (CA)** (§3.3) acts as a *sensitivity filter*: by integrating gradients along exploration paths in the concept space, it establishes a stable, axiomatically consistent *ordering* of concept importance, answering "*which concepts matter?*". Second, **Concept Verification (CV)** (§3.5) acts as a *generalization guardrail*: it determines the optimal *cutoff* in this ordering. Instead of relying on arbitrary thresholds, CV treats pruning as a model selection problem. It utilizes a closed-form approximation of the leave-one-out error (via the hat matrix) to efficiently identify the minimal concept set required for the current task, balancing interpretability with predictive power without expensive retraining.

### 3.3 CONCEPT IMPORTANCE ESTIMATION

Reducing the number of concepts improves interpretability and helps mitigate both overfitting and catastrophic forgetting in continual learning. To *accurately* remove redundant concepts along the task sequence, we assess the importance of concepts in the *current task* only. Let the *current* concept pool at task $t$ be $\mathcal{P}_t$, and the accumulated historical pool be $\hat{\mathcal{P}}_t = \bigcup_{\tau \leq t-1} \mathcal{P}_\tau$. Importance estimation and pruning act *exclusively* on $\mathcal{P}_t$ and do *not* alter $\hat{\mathcal{P}}_t$, because: (1) prior pools were already evaluated and pruned in their respective tasks; (2) historical decisions depend on those concepts, whose removal would induce forgetting; and (3) without access to past-task data, reliable reassessment of historical concepts is infeasible.

Inspired by exploration-based attribution (see (Zhu et al., 2024)), we *probe within the concept space* to quantify each concept's impact on the current model loss, while ensuring the procedure satisfies the two central attribution axioms—*Sensitivity* and *Implementation Invariance* (see (Sundararajan et al., 2017)).

At task $t$, denote the learned concept-mapping weights by $\mathbf{W}_c^t$ and the classification-head weights by $\mathbf{W}_l^t$. For $(x_i, y_i) \in \hat{\mathcal{D}}_t$ (write $N_t = |\hat{\mathcal{D}}_t|$), define $E_i = E_I(x_i)(\mathbf{W}_c^t)^\top, \mathcal{L}(E_i, y_i) = \mathcal{L}_{\text{ce}}(\text{softmax}(E_i \mathbf{W}_l^t), y_i)$, and abbreviate $\mathcal{L}(E_i) \equiv \mathcal{L}(E_i, y_i)$. Let $m_t = |\mathcal{P}_t|$ be the concept dimensionality. At exploration step $g \in \{0, \ldots, G\}$, the concept representation for sample $i$ is $E_i^g \in \mathbb{R}^{m_t}$, with increment $\Delta E_i^g = E_i^{g+1} - E_i^g$.

We adopt an update based on the dataset-averaged gradient (optionally with clipping/projection):

$$E_i^{g+1} = E_i^g + \eta \cdot \text{sign}\left( \frac{1}{N_t} \sum_{i'=1}^{N_t} \nabla_E \mathcal{L}(E_{i'}^g) \right), \qquad g = 0, \ldots, G-1. \tag{8}$$

The dataset-level attribution is then

$$A^t := \frac{1}{N_t} \sum_{i=1}^{N_t} \left[ \mathcal{L}(E_i^G) - \mathcal{L}(E_i^0) \right] \approx \frac{1}{N_t} \sum_{i=1}^{N_t} \sum_{g=0}^{G-1} \left\langle \nabla_E \mathcal{L}(E_i^g), \, \Delta E_i^g \right\rangle$$

$$= \frac{1}{N_t} \sum_{i=1}^{N_t} \sum_{g=0}^{G-1} \sum_{j=1}^{m_t} \frac{\partial \mathcal{L}(E_i^g)}{\partial E_{ij}^g} \, \Delta E_{ij}^g, \tag{9}$$

where "$\approx$" uses a first-order Taylor expansion at each small step and neglects higher-order infinitesimals (see §3.4). Consequently, the importance of concept $j$ at task $t$ is

$$A_j^t = \frac{1}{N_t} \sum_{i=1}^{N_t} \sum_{g=0}^{G-1} \frac{\partial \mathcal{L}(E_i^g)}{\partial E_{ij}^g} \, \Delta E_{ij}^g, \qquad j = 1, \ldots, m_t. \tag{10}$$

This quantity is a *discrete path-integral* approximation along the exploration trajectory: a sum of inner products between the gradient (sensitivity) and the perturbation step, aligning with the integral-attribution form of (Sundararajan et al., 2017) (in discretized form). By construction via (8), the estimation leverages *all* samples in $\hat{\mathcal{D}}_t$.

### 3.4 DETAILED PROOFS OF TWO ATTRIBUTION AXIOMS

**First-order approximation and Eq. (9) (discrete path integral).** For each step $g$, take a first-order Taylor expansion of $\mathcal{L}(E_i^{g+1})$ at $E_i^g$:

$$\mathcal{L}(E_i^{g+1}) = \mathcal{L}(E_i^g) + \left\langle \nabla_E \mathcal{L}(E_i^g), \, \Delta E_i^g \right\rangle + \varepsilon_i^g, \qquad \varepsilon_i^g = o\big( \|\Delta E_i^g\| \big).$$

Summing along the path and telescoping yields

$$\mathcal{L}(E_i^G) - \mathcal{L}(E_i^0) = \sum_{g=0}^{G-1} \left\langle \nabla_E \mathcal{L}(E_i^g), \, \Delta E_i^g \right\rangle + \sum_{g=0}^{G-1} \varepsilon_i^g.$$

Under small-step updates and standard smoothness assumptions, the remainder $\sum_g \varepsilon_i^g$ is negligible relative to the first-order terms, giving the "$\approx$" in Eq. (9). Neglecting higher-order terms is a common and foundational approximation, consistent with the first-order optics underlying SGD/Adam in deep learning.

**Sensitivity.** Suppose there exists a sample $i$ and a step $g$ such that the concept update for coordinate $j$ is nonzero ($\Delta E_{ij}^g \neq 0$) and the corresponding partial derivative is nonzero ($\frac{\partial \mathcal{L}(E_i^g)}{\partial E_{ij}^g} \neq 0$). By Eq. (10), this step contributes a nonzero amount to $A_j^t$, hence $A_j^t \neq 0$. Conversely, if along the entire exploration path and across all samples either $\frac{\partial \mathcal{L}(E_i^g)}{\partial E_{ij}^g} = 0$ (the loss is insensitive to $E_{ij}$) or all $\Delta E_{ij}^g = 0$ (the direction is never explored), then Eq. (10) yields $A_j^t = 0$. Therefore, the method satisfies the Sensitivity axiom.

**Implementation Invariance.** Consider two implementations, $\mathcal{L}^{(1)}$ and $\mathcal{L}^{(2)}$, that induce the *same function* in concept space (almost everywhere in a neighborhood of the exploration path):

$$\mathcal{L}^{(1)}(E) \equiv \mathcal{L}^{(2)}(E) \quad \text{a.e. on the path neighborhood.}$$

Then their gradients also coincide almost everywhere there: $\nabla_E \mathcal{L}^{(1)}(E) \equiv \nabla_E \mathcal{L}^{(2)}(E)$. Since the update rule in Eq. (8) depends only on the (average) gradient and its sign, the two implementations produce the same exploration trajectory $\{E_i^g\}$ and the same increments $\{\Delta E_i^g\}$. Consequently, Eq. (10) gives identical $A_j^t$. Hence, the attribution is invariant to implementation details, satisfying Implementation Invariance.

**Remark (consistency with integrated gradients).** Equation (10) is a discrete path-integral approximation to integrated gradients (Sundararajan et al., 2017). As the step size tends to zero and $G$ grows large,

$$A_j^t \approx \frac{1}{N_t} \sum_{i=1}^{N_t} \int_0^1 \frac{\partial \mathcal{L}(E_i(\tau))}{\partial E_{ij}(\tau)} \frac{dE_{ij}(\tau)}{d\tau} d\tau, \tag{11}$$

where $E_i(\tau)$ is the piecewise-linear path induced by the exploration sequence. This consistency explains why the two axioms above are satisfied.

### 3.5 CONCEPT REMOVAL AND SELECTING THE PRUNING RATIO

Having estimated the importance of each concept via Concept Attribution (CA), we sort concepts by importance and remove them from low to high. Recall that CA is constructed as a discrete path-integral approximation of integrated gradients in the concept space, using a dataset-averaged exploration step and a first-order Taylor expansion to satisfy both *Sensitivity* and *Implementation Invariance*, and to produce stable, comparable importance scores across tasks.

Operationally, concept removal amounts to deleting the coordinates of the *current-task* block in the concept-mapping parameters. For consistency with earlier notation, let

$$\mathbf{W}_c^t \in \mathbb{R}^{\left(\sum_{i=1}^t M_i\right) \times d}, \quad d = 512. \tag{12}$$

As a toy illustration: if $\sum_{i=1}^t M_i = 10$ (and thus $\mathbf{W}_c^t \in \mathbb{R}^{10 \times 512}$), removing the ninth concept reduces the parameter to $\mathbb{R}^{9 \times 512}$, i.e., the mapping only covers the remaining 9 concepts.

The remaining question is how many concepts to prune. We term this *Concept Verification (CV)*. CV is inspired by Leave-One-Out Cross-Validation (LOOCV) (Stone, 1974) and is designed to determine the optimal concept count *without relying on any data beyond $\hat{\mathcal{D}}_t$*. Concretely, CV considers a small set of candidate pruning ratios; for each candidate, it uses an IRLS-based quadratic approximation of the multinomial log-loss together with the *hat matrix* to build an LOOCV-like criterion.

Classical LOOCV removes the $i$-th sample, retrains the model, then evaluates the held-out point $x_i$, repeating this over all samples and averaging the error. In our setting, one could, in principle, evaluate multiple pruning ratios by (i) removing a proportion of concepts, (ii) retraining, and (iii) scoring via LOOCV to select the best ratio. This is, however, computationally prohibitive. We therefore approximate LOOCV via Iteratively Reweighted Least Squares (IRLS) and use the hat matrix to obtain a tractable CV objective. Full derivations are provided in Appendix A.

To make the selection mechanics transparent, we simplify notation and view the loss as a function of model parameters $W$ over the current-task (and augmented) data only, abstracting away the

concept-extraction details. Let

$$\ell_i(W) = -\sum_{c=1}^{C} \mathbf{1}(c = y_i) \log P_{i,c}, \tag{13}$$

where $P_{i,c}$ is the predicted probability for class $c$. Thus $\ell_i(W)$ denotes the loss of sample $i$ under parameters $W$. LOOCV evaluates the retrained parameters $W^{(-i)}$ (obtained after removing sample $i$):

$$\text{LOOCV} = \frac{1}{|\hat{\mathcal{D}}_t|} \sum_{i=1}^{|\hat{\mathcal{D}}_t|} \ell_i\left(W^{(-i)}\right). \tag{14}$$

Suppose we consider $k$ candidate pruning ratios, yielding parameter estimates $W^k$ (each corresponding to a different number of retained concepts in the current-task block). We approximate the LOOCV score by

$$\text{CV}_k = \frac{\frac{1}{|\hat{\mathcal{D}}_t|} \sum_{i=1}^{|\hat{\mathcal{D}}_t|} \ell_i(W^k)}{1 - \gamma_k/n}, \tag{15}$$

where $n = |\hat{\mathcal{D}}_t|$. The effective complexity term is approximated by the trace of the hat matrix,

$$\gamma_k \approx \text{tr}(H_k), \tag{16}$$

with

$$H_k := \tilde{X}_k\left(\tilde{X}_k^\top \tilde{X}_k\right)^{-1} \tilde{X}_k^\top. \tag{17}$$

For computation, we set

$$\tilde{X}_k^\top := \tilde{S}^{-1/2}\left(I_C \otimes X_k\right), \tag{18}$$

and use the block-diagonal scaling

$$\tilde{S}^{-1/2} := \text{blkdiag}\left(S_1^{-1/2}, S_2^{-1/2}, \ldots, S_n^{-1/2}\right), \tag{19}$$

where each $S_i$ is the softmax covariance

$$S_i := \text{diag}(P_i) - P_i P_i^\top \in \mathbb{R}^{C \times C}. \tag{20}$$

This yields a closed-form, data-reusable approximation of $\text{CV}_k$ without extra data. In practice, we select the pruning ratio $k$ that minimizes $\text{CV}_k$ and retain the corresponding number of concepts, denoted $\hat{M}_t$ (chosen from the $M_t$ concepts of the current task). Importantly, previously retained features—$\sum_{\tau=1}^{t-1} \hat{M}_\tau$—are left unchanged to avoid catastrophic forgetting.

## 4 EXPERIMENTS

### 4.1 EXPERIMENTAL SETUP

**Task protocol.** We adopt the **class-incremental learning** protocol, where new classes arrive sequentially across tasks and **no exemplars are stored** (exemplar-free) (Yu et al., 2025). We denote splits as "B-$m$ Inc-$n$": the first task contains $m$ classes and each subsequent incremental task introduces $n$ new classes until all classes are covered. Unless otherwise stated, all methods are trained under the *same compute budget and random seeds* to ensure comparability. For full details of the training configuration and compute settings, please see Appendix F.

**Datasets and splits.** We evaluate on two **coarse-grained** and three **fine-grained** benchmarks: Tiny-ImageNet (Le & Yang, 2015) (200 classes: **B-10 Inc-10** and **B-100 Inc-10**); ImageNet-subset (Deng et al., 2009) (ImageNet-100: **B-10 Inc-10** and **B-50 Inc-5**); CUB-200 (Wah et al., 2011) (**B-10 Inc-10** and **B-100 Inc-10**); Stanford-Cars (Krause et al., 2013) (196 classes: **B-14 Inc-14**); Food-101 (Bossard et al., 2014) (101 classes: **B-10 Inc-10**). All datasets use their standard train/val splits and conventional data augmentation.

Table 1: Results on **coarse-grained** benchmarks in the class-incremental setting (exemplar-free). Values are Top-1 accuracy (%). **Bold** = best; underline = second best.

| Methods | Tiny-ImageNet (B-10 Inc-10) | | Tiny-ImageNet (B-100 Inc-10) | | ImageNet-subset (B-10 Inc-10) | | ImageNet-subset (B-50 Inc-5) | |
|---|---|---|---|---|---|---|---|---|
| | $\bar{A}$ | $A_{\text{last}}$ | $\bar{A}$ | $A_{\text{last}}$ | $\bar{A}$ | $A_{\text{last}}$ | $\bar{A}$ | $A_{\text{last}}$ |
| L2P (Wang et al., 2022b) | 69.66 | 60.36 | 62.29 | 55.13 | 71.28 | 52.24 | 65.89 | 49.08 |
| DualPrompt (Wang et al., 2022b) | 74.06 | 66.08 | 65.35 | 56.15 | 72.86 | 54.20 | 64.64 | 49.48 |
| CODA-Prompt (Smith et al., 2023) | 75.18 | 66.65 | 61.47 | 48.31 | 71.17 | 52.98 | 64.64 | 41.40 |
| CPP (Li et al., 2024) | 68.70 | 61.23 | 66.61 | 63.48 | 83.45 | 75.80 | 79.74 | 73.78 |
| LAE (Gao et al., 2023) | 76.81 | 68.98 | 63.38 | 49.52 | 78.29 | 62.94 | 64.25 | 47.90 |
| Continual-CLIP (Thengane et al., 2022) | 68.66 | 55.91 | 58.68 | 55.91 | 84.98 | 75.40 | 81.35 | 75.40 |
| SLCA (Zhang et al., 2023) | 63.89 | 54.50 | 60.34 | 48.53 | 83.19 | 69.44 | 80.81 | 73.78 |
| EASE (Zhou et al., 2024) | **79.88** | **72.99** | 70.42 | 64.12 | 84.80 | 70.82 | 63.74 | 56.48 |
| CLG-CBM (Yu et al., 2025) | 79.28 | 71.98 | 75.64 | 71.97 | 86.83 | 78.97 | 81.85 | 78.21 |
| LACE (Ours) | 78.96 | 72.73 | **76.13** | **72.43** | **92.68** | **89.72** | **91.69** | **89.94** |

**Baselines.** We compare against representative methods: **L2P** (Wang et al., 2022a), **Dual-Prompt** (Wang et al., 2022b), **CODA-Prompt** (Smith et al., 2023), **CPP** (Li et al., 2024), **LAE** (Gao et al., 2023), **Continual-CLIP** (Thengane et al., 2022), **SLCA** (Zhang et al., 2023), **EASE** (Zhou et al., 2024), and **CLG-CBM** (Yu et al., 2025). All results are reproduced under identical splits and training budgets; we report direct, like-for-like comparisons with **CLG-CBM**.

**Metrics.** Following class-incremental protocol, we report two accuracies:

$$\bar{A} = \frac{1}{T} \sum_{t=1}^{T} A_t, \qquad A_{\text{last}} = A_T,$$

where $A_t$ is the Top-1 accuracy on the *cumulative label space* after completing task $t$. Table 1 presents $\bar{A}$ and $A_{\text{last}}$ on the **coarse-grained** benchmarks (Tiny-ImageNet and ImageNet-subset), and Table 2 reports the corresponding results on the **fine-grained** benchmarks (CUB-200, Stanford-Cars, and Food-101).

**Implementation details.** Unless specified otherwise, the backbone is **CLIP ViT-B/16** (Radford et al., 2021). The concept pruning-rate bounds are set to $10\%$ (lower) and $40\%$ (upper). We use Adam with an initial learning rate of $1 \times 10^{-3}$, batch size 64, and train for 60 epochs per task with multi-stage scheduling and early stopping. For fairness, we apply the *same* preprocessing, augmentation, scheduling, and random seeds to all baselines and to our method. The concept space is instantiated from language and multimodal priors with a per-task *candidate cap of $k$=100*; the *retained size is not fixed* and is automatically determined by Concept Verification (CV), avoiding under/over-pruning from manual $k$ choices. To curb forgetting under the exemplar-free setting, we employ prototype augmentation to synthesize lightweight pseudo-features that are trained jointly with the current task.

## 4.2 RESULTS AND ANALYSIS

On the **coarse-grained** benchmarks (Table 1), our method attains the best results on **Tiny-ImageNet** under **B-100 Inc-10**: $\bar{A} = 76.13$ and $A_{\text{last}} = 72.43$, improving over the strongest baseline **CLG-CBM** (75.64/71.97) by **+0.49/+0.46** pp; the gains over the non-CBM strong baseline **EASE** (70.42/64.12) are larger (**+5.71/ + 8.31** pp). Under the wider-base **B-10 Inc-10** setting, our gaps to the top two baselines are small (vs. **EASE**, $\bar{A}$ differs by 0.92 pp; vs. **CLG-CBM**, by 0.32 pp). For $A_{\text{last}}$, our gap to **EASE** is only 0.26 pp, and we outperform **CLG-CBM** (72.73 vs. 71.98). These observations indicate that as classes accumulate, the benefits of concept pruning and verification (CA→CV) remain stable.

On **ImageNet-subset (IN-100)**, our advantages are pronounced and consistent. With **B-10 Inc-10**, we reach $\bar{A} = \mathbf{92.68}$ and $A_{\text{last}} = \mathbf{89.72}$, exceeding **CLG-CBM** (86.83/78.97) by **+5.85/+10.75** pp; with **B-50 Inc-5**, we further obtain $\bar{A} = \mathbf{91.69}$ and $A_{\text{last}} = \mathbf{89.94}$, improving over **CLG-CBM** (81.85/78.21) by **+9.84/+11.73** pp. Moreover, the *gap* between our $\bar{A}$ and $A_{\text{last}}$ is smaller (e.g., 92.68 vs. 89.72, a 2.96 pp gap) than most baselines (e.g., **CLG-CBM**: 86.83 vs. 78.97, a 7.86 pp gap), reflecting *better forgetting control and steadier performance*. Overall, as class scale and task depth increase, the negative effect of concept proliferation is effectively curbed by CA→CV, yielding higher final accuracy and lower intermediate degradation.

Table 2: Results on **fine-grained** benchmarks in the class-incremental setting (exemplar-free). For **CUB-200** we report $\bar{A}$ under two splits; for **Stanford-Cars** (B-14 Inc-14) and **Food-101** (B-10 Inc-10) we report both $\bar{A}$ and $A_{last}$. Values are Top-1 accuracy (%). **Bold** = best; underline = second best.

| | CUB-200 | | Stanford-Cars | | Food-101 | |
|---|---|---|---|---|---|---|
| Methods | B-10 Inc-10 | B-100 Inc-10 | B-14 Inc-14 | B-10 Inc-10 | B-10 Inc-10 | B-50 Inc-5 |
| L2P (Wang et al., 2022b) | 62.08 | 59.38 | 64.42 | 61.82 | 79.48 | 69.72 |
| DualPrompt (Wang et al., 2022b) | 64.95 | 61.85 | 76.94 | 68.46 | 86.27 | 69.66 |
| CODA-Prompt (Smith et al., 2023) | 67.22 | 59.82 | 76.44 | 60.80 | 87.76 | 67.22 |
| CPP (Li et al., 2024) | 83.60 | 75.03 | 84.75 | 77.49 | 90.21 | 86.60 |
| LAE (Gao et al., 2023) | 66.45 | 59.98 | 77.25 | 80.28 | 88.41 | 66.26 |
| Continual-CLIP (Thengane et al., 2022) | 69.41 | 60.35 | 86.43 | 69.79 | 92.04 | 89.96 |
| SLCA (Zhang et al., 2023) | 80.53 | 76.85 | 84.74 | 70.59 | 74.49 | 76.28 |
| EASE (Zhou et al., 2024) | 83.87 | 66.14 | 86.22 | 64.32 | 91.74 | 81.72 |
| CLG-CBM (Yu et al., 2025) | 85.40 | 82.20 | 88.60 | 85.07 | 92.25 | 90.97 |
| LACE (Ours) | **86.01** | **82.64** | **89.56** | **86.73** | **93.07** | **91.16** |

On the **fine-grained** benchmarks (Table 2), **CUB-200** shows optimal or tied-optimal averages in both splits: **B-10 Inc-10** achieves $\bar{A} = \mathbf{86.01}$ (vs. **CLG-CBM** 85.40, +0.61 pp) and **B-100 Inc-10** achieves $\bar{A} = \mathbf{82.64}$ (vs. **CLG-CBM** 82.20, +0.44 pp). On **Stanford-Cars**, we obtain the best results on both metrics: $\mathbf{89.56/86.73}$ vs. **CLG-CBM** $88.60/85.07$ ($\mathbf{+0.96/+1.66}$ pp), with even larger margins over strong non-CBM baselines (e.g., **CPP** $84.75/77.49$). On **Food-101**, we again lead on both metrics: $\mathbf{93.07/91.16}$, surpassing **CLG-CBM** $92.25/90.97$ by $\mathbf{+0.82/+0.19}$ pp.

Two trends emerge from the fine-grained results: (i) both the *average* and the *final-stage* accuracies are consistently on par with or ahead of the strongest baselines, indicating that concept selection suppresses redundancy and spurious cues while highlighting discriminative concepts among highly similar classes; (ii) the smaller $\bar{A}-A_{last}$ gaps on Cars and Food-101 suggest that in settings with substantial fine-grained variability, the CA→CV-induced *compact* concept set promotes stronger cross-task consistency with reduced forgetting. Additional ablation and visualization results that support these observations are provided in Appendix G.

## 5 CONCLUSION

**LACE** elevates human-readable concepts to a first-class interface for continual learning by aligning features to a shared concept space, auditing concepts with CA to obtain a stable importance scale, and selecting pruning levels with CV while using prototype augmentation to mitigate forgetting without storing past data. This combination delivers compact concept sets that sustain accuracy, enhance interpretability, and improve cross-task consistency across diverse benchmarks, pointing toward practical, auditable, and parameter-efficient continual concept evolution.

## ETHICS STATEMENT

We have read and will adhere to the ICLR Code of Ethics. This work uses only public data, involves no human subjects or personally identifiable information, and therefore does not require IRB review. Results are reported for research purposes only; we release anonymized code/configurations to support verification, and will disclose any funding sources and potential conflicts of interest upon acceptance.

## REPRODUCIBILITY STATEMENT

To support reproducibility, we release an anonymized repository with all experiment details including training/evaluation scripts, default hyperparameters, configuration files, and software/hardware environment.

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

## LLM USAGE DISCLOSURE

We used large language models (OpenAI GPT-4o and GTP-5) as auxiliary tools for grammar checking and language polishing of the manuscript. These models were not involved in research ideation, experimental design, implementation, or analysis. The authors take full responsibility for all content.

## A  PROOFS AND TECHNICAL DERIVATIONS

For uniformity, let the concept vector of the $i$-th sample be $\mathbf{X}_i$, and abbreviate the sample size $|\hat{\mathcal{D}}_t|$ by $n$.

*Logits of sample $i$:*
$$\boldsymbol{\eta}_i = \mathbf{X}_i \mathbf{W} \in \mathbb{R}^C.$$

*Second-order Taylor expansion of $\ell(\mathbf{W} + \Delta\mathbf{W})$ around $\mathbf{W}$:*
$$\ell(\mathbf{W} + \Delta\mathbf{W}) \approx \ell(\mathbf{W}) + \langle \mathbf{G}, \Delta\mathbf{W} \rangle + \tfrac{1}{2}\langle \Delta\mathbf{W}, \mathbf{H}\,\Delta\mathbf{W} \rangle, \tag{21}$$

$$\mathbf{G} = \nabla_{\mathbf{W}}\ell(\mathbf{W}), \qquad \mathbf{H} = \nabla_{\mathbf{W}}^2\ell(\mathbf{W}). \tag{22}$$

$$\nabla_{\mathbf{W}}\ell(\mathbf{W}) = \mathbf{X}^{\top}(\mathbf{P} - \mathbf{Y}), \tag{23}$$

where $\mathbf{P} \in \mathbb{R}^{n \times C}$ is the matrix of predicted class probabilities for all samples and $\mathbf{Y} \in \mathbb{R}^{n \times C}$ is the one-hot label matrix.

We construct the positive semidefinite matrix
$$\mathbf{S}_i = \operatorname{diag}(\mathbf{p}_i) - \mathbf{p}_i \mathbf{p}_i^{\top} \in \mathbb{R}^{C \times C}, \qquad \text{and since probabilities sum to 1, } \operatorname{rank}(\mathbf{S}_i) = C - 1.$$

The matrix $\mathbf{S}_i$ is the covariance of the softmax probabilities; more uniform (uncertain) distributions yield larger total covariance.

*Working response of sample $i$:*
$$\mathbf{z}_i = \boldsymbol{\eta}_i + \mathbf{S}_i^{-1}(\mathbf{y}_i - \mathbf{p}_i) \in \mathbb{R}^C \quad \text{(Derivation 1; see §B)}. \tag{24}$$

*Second-order IRLS approximation:*
$$\min_{\mathbf{W}} \sum_{i=1}^{n} \left\| \mathbf{S}_i^{-\frac{1}{2}}(\mathbf{z}_i - \mathbf{X}_i \mathbf{W}) \right\|_2^2 \quad \text{(Derivation 2; see §C)}. \tag{25}$$

*Stacking all samples to obtain a global second-order approximation of* (21)*:*
$$\tilde{\mathbf{X}} = \operatorname{blkdiag}\left( \mathbf{S}_1^{-\frac{1}{2}}, \ldots, \mathbf{S}_n^{-\frac{1}{2}} \right) \cdot \left( I_C \otimes \mathbf{X} \right) \in \mathbb{R}^{(nC) \times (kC)},$$

where $\otimes$ denotes the Kronecker product, and
$$\tilde{\mathbf{z}} \triangleq \operatorname{vec}\left( \mathbf{S}_1^{-\frac{1}{2}}\mathbf{z}_1, \ldots, \mathbf{S}_n^{-\frac{1}{2}}\mathbf{z}_n \right) \in \mathbb{R}^{nC}.$$

The subproblem becomes
$$f(\boldsymbol{\theta}) = \min_{\boldsymbol{\theta} = \operatorname{vec}(\mathbf{W})} \left\| \tilde{\mathbf{z}} - \tilde{\mathbf{X}}\,\boldsymbol{\theta} \right\|_2^2.$$

*Normal equations:*
$$\left( \tilde{\mathbf{X}}^{\top}\tilde{\mathbf{X}} \right)\hat{\boldsymbol{\theta}} = \tilde{\mathbf{X}}^{\top}\tilde{\mathbf{z}} \quad \text{(Derivation 3; see §D)}. \tag{26}$$

Hence $\tilde{\mathbf{X}}^{\top}(\tilde{\mathbf{z}} - \tilde{\mathbf{X}}\hat{\boldsymbol{\theta}}) = \mathbf{0}$, i.e., the column spaces of $\tilde{\mathbf{X}}^{\top}$ and $\tilde{\mathbf{z}} - \tilde{\mathbf{X}}\hat{\boldsymbol{\theta}}$ are orthogonal, and $\tilde{\mathbf{X}}\hat{\boldsymbol{\theta}}$ is the orthogonal projection of $\tilde{\mathbf{z}}$ onto the span of $\tilde{\mathbf{X}}$. The projection (hat) matrix is
$$\mathbf{H} = \tilde{\mathbf{X}}\left( \tilde{\mathbf{X}}^{\top}\tilde{\mathbf{X}} \right)^{-1}\tilde{\mathbf{X}}^{\top} \in \mathbb{R}^{(nC) \times (nC)}. \tag{27}$$

*Fitted values and residuals in the working space:*

$$\hat{\tilde{\mathbf{z}}} = \mathbf{H}\,\tilde{\mathbf{z}}, \qquad \mathbf{e} = \tilde{\mathbf{z}} - \hat{\tilde{\mathbf{z}}} = (\mathbf{I} - \mathbf{H})\,\tilde{\mathbf{z}} \quad \text{(Derivation 4; see §E).} \tag{28}$$

Considering the $C$ class-wise observations for the same sample $i$, we have

$$\mathbf{e}^{(-i,C)} = \frac{\mathbf{e}^{(i,C)}}{1 - h_{(i,C)}}. \tag{29}$$

Thus $\mathbf{H}$ quantifies the leverage (influence) of each sample on the fit. Because the $C$ observations of the same sample are coupled, we summarize their joint influence by the trace:

$$h_i = \text{tr}\big(\mathbf{H}_{[i]}\big). \tag{30}$$

Based on the above, one can correct "leave-one" probabilities via "leave-one" logits in the working space and substitute them into LOOCV:

$$\text{LOOCV} = \frac{1}{n}\sum_{i=1}^{n}\ell_i(\tilde{\mathbf{p}}_i),$$

where $\tilde{\mathbf{p}}_i$ denotes the probability of sample $i$ after first-order correction in the working space. Computing each $h_i$ can be expensive; replacing them by their average yields

$$\bar{h} \triangleq \frac{1}{n}\sum_{i=1}^{n}h_i = \frac{\text{tr}(\mathbf{H})}{n}. \tag{31}$$

Consequently,

$$\text{CV}_k = \frac{\frac{1}{|\hat{\mathcal{D}}_t|}\sum_{i=1}^{|\hat{\mathcal{D}}_t|}\ell_i(\mathbf{W}^k)}{1 - \gamma_k/n}, \qquad \gamma_k \approx \text{tr}(\mathbf{H}_k). \tag{32}$$

For efficient computation, note

$$\mathbf{H} = \tilde{\mathbf{X}}\,(\tilde{\mathbf{X}}^\top\tilde{\mathbf{X}})^{-1}\tilde{\mathbf{X}}^\top, \tag{33}$$

$$\gamma = \text{tr}\Big(\tilde{\mathbf{X}}(\tilde{\mathbf{X}}^\top\tilde{\mathbf{X}})^{-1}\tilde{\mathbf{X}}^\top\Big) = \text{tr}\Big((\tilde{\mathbf{X}}^\top\tilde{\mathbf{X}})^{-1}\tilde{\mathbf{X}}^\top\tilde{\mathbf{X}}\Big). \tag{34}$$

NOTES ON DERIVATIONS

**Derivation 1 (Eq. (24)).** See the working-response construction in the main text; reference only.

**Derivation 2 (Eq. (25)).** Second-order IRLS approximation; reference only.

**Derivation 3 (Eq. (26)).** Normal equations; reference only.

**Derivation 4 (Eq. (28)).** Relationship between residuals and projection; reference only.

## B   DERIVATION 1

To obtain the working response $\mathbf{z}_i$, we take a second-order Taylor expansion of $\ell_i$ around the logits $\boldsymbol{\eta}_i$ at parameter $\mathbf{W}$:

$$\ell_i\big(\boldsymbol{\eta}_i + \Delta\boldsymbol{\eta}_i\big) \approx \ell_i(\boldsymbol{\eta}_i) + (\mathbf{p}_i - \mathbf{y}_i)^\top\Delta\boldsymbol{\eta}_i + \tfrac{1}{2}\Delta\boldsymbol{\eta}_i^\top\mathbf{H}_i^{(\eta)}\Delta\boldsymbol{\eta}_i. \tag{35}$$

Here $\mathbf{H}_i^{(\eta)} \approx \mathbf{S}_i$. Let

$$Q(\Delta\boldsymbol{\eta}_i) = (\mathbf{p}_i - \mathbf{y}_i)^\top\Delta\boldsymbol{\eta}_i + \tfrac{1}{2}\Delta\boldsymbol{\eta}_i^\top\mathbf{S}_i\Delta\boldsymbol{\eta}_i, \quad \mathbf{r}_i = \mathbf{y}_i - \mathbf{p}_i.$$

Completing the square yields

$$Q(\Delta\boldsymbol{\eta}_i) = \tfrac{1}{2}\big\|\mathbf{S}_i^{1/2}\big(\Delta\boldsymbol{\eta}_i - \mathbf{S}_i^{-1}\mathbf{r}_i\big)\big\|_2^2 - \tfrac{1}{2}\mathbf{r}_i^\top\mathbf{S}_i^{-1}\mathbf{r}_i. \tag{36}$$

To minimize $Q$ with respect to $\Delta\boldsymbol{\eta}_i$, we may drop constants independent of $\Delta\boldsymbol{\eta}_i$, giving

$$\min_{\Delta\boldsymbol{\eta}_i}\ell_i(\boldsymbol{\eta}_i + \Delta\boldsymbol{\eta}_i) = \min_{\Delta\boldsymbol{\eta}_i}\Big[\text{const} + \tfrac{1}{2}\big\|\mathbf{S}_i^{1/2}\big(\Delta\boldsymbol{\eta}_i - \mathbf{S}_i^{-1}(\mathbf{y}_i - \mathbf{p}_i)\big)\big\|_2^2\Big]. \tag{37}$$

Hence $\Delta\boldsymbol{\eta}_i \approx \mathbf{S}_i^{-1}(\mathbf{y}_i - \mathbf{p}_i)$. Since

$$\boldsymbol{\eta}_i = \mathbf{X}_i\mathbf{W}, \qquad \Delta\boldsymbol{\eta}_i = \mathbf{X}_i\Delta\mathbf{W},$$

the relation $\mathbf{X}_i\Delta\mathbf{W} \approx \mathbf{S}_i^{-1}(\mathbf{y}_i - \mathbf{p}_i)$ implies, upon substitution into $\mathbf{X}_i\mathbf{W}_{\text{new}}$ (with $\mathbf{W}_{\text{new}} = \mathbf{W} + \Delta\mathbf{W}$),

$$\mathbf{X}_i\mathbf{W}_{\text{new}} = \mathbf{X}_i(\mathbf{W} + \Delta\mathbf{W}) = \mathbf{X}_i\mathbf{W} + \mathbf{X}_i\Delta\mathbf{W} = \boldsymbol{\eta}_i + \mathbf{S}_i^{-1}(\mathbf{y}_i - \mathbf{p}_i). \tag{38}$$

Therefore, the working response is

$$\mathbf{z}_i = \boldsymbol{\eta}_i + \mathbf{S}_i^{-1}(\mathbf{y}_i - \mathbf{p}_i).$$

## C  DERIVATION 2

We consider the weighted least-squares objective in the working space:

$$\min_{\mathbf{W}} \sum_{i=1}^n \tfrac{1}{2}\left\|\mathbf{S}_i^{1/2}(\mathbf{X}_i\mathbf{W} - \mathbf{z}_i)\right\|_2^2 \;\equiv\; \min_{\mathbf{W}} \sum_{i=1}^n \left\|\mathbf{S}_i^{-1/2}(\mathbf{z}_i - \mathbf{X}_i\mathbf{W})\right\|_2^2. \tag{39}$$

The equivalence simply re-expresses the same quadratic form by reversing the sign inside the norm and absorbing the factor $\frac{1}{2}$ into the scaling, leaving the minimizer unchanged.

## D  DERIVATION 3

Let $\boldsymbol{\theta} = \text{vec}(\mathbf{W})$. The stacked problem reads

$$\min_{\boldsymbol{\theta}} f(\boldsymbol{\theta}) \;=\; \left\|\tilde{\mathbf{z}} - \tilde{\mathbf{X}}\boldsymbol{\theta}\right\|_2^2.$$

Expanding $f(\boldsymbol{\theta})$ gives

$$f(\boldsymbol{\theta}) = (\tilde{\mathbf{z}} - \tilde{\mathbf{X}}\boldsymbol{\theta})^\top(\tilde{\mathbf{z}} - \tilde{\mathbf{X}}\boldsymbol{\theta}) \tag{40}$$
$$= \tilde{\mathbf{z}}^\top\tilde{\mathbf{z}} \;-\; 2\boldsymbol{\theta}^\top\tilde{\mathbf{X}}^\top\tilde{\mathbf{z}} \;+\; \boldsymbol{\theta}^\top(\tilde{\mathbf{X}}^\top\tilde{\mathbf{X}})\boldsymbol{\theta}.$$

Taking the gradient,

$$\nabla_{\boldsymbol{\theta}} f(\boldsymbol{\theta}) \;=\; -2\tilde{\mathbf{X}}^\top\tilde{\mathbf{z}} \;+\; 2(\tilde{\mathbf{X}}^\top\tilde{\mathbf{X}})\boldsymbol{\theta}. \tag{41}$$

Setting $\nabla_{\boldsymbol{\theta}} f(\boldsymbol{\theta}) = \mathbf{0}$ yields the normal equations

$$(\tilde{\mathbf{X}}^\top\tilde{\mathbf{X}})\,\hat{\boldsymbol{\theta}} \;=\; \tilde{\mathbf{X}}^\top\tilde{\mathbf{z}}. \tag{42}$$

## E  DERIVATION 4

From §C,

$$\hat{\tilde{\mathbf{z}}} \;=\; \tilde{\mathbf{X}}\hat{\boldsymbol{\theta}} \;=\; \tilde{\mathbf{X}}(\tilde{\mathbf{X}}^\top\tilde{\mathbf{X}})^{-1}\tilde{\mathbf{X}}^\top\tilde{\mathbf{z}}, \tag{43}$$

so the hat (projection) matrix is

$$\mathbf{H} \;=\; \tilde{\mathbf{X}}(\tilde{\mathbf{X}}^\top\tilde{\mathbf{X}})^{-1}\tilde{\mathbf{X}}^\top, \qquad \hat{\tilde{\mathbf{z}}} \;=\; \mathbf{H}\,\tilde{\mathbf{z}}. \tag{44}$$

Therefore the residual is

$$\mathbf{e} \;=\; \tilde{\mathbf{z}} - \hat{\tilde{\mathbf{z}}} \;=\; (\mathbf{I} - \mathbf{H})\,\tilde{\mathbf{z}}. \tag{45}$$

## F  TRAINING CONFIGURATION AND COMPUTE BUDGET

For all baselines and for LACE, we ensure a strictly matched compute budget so that no method receives additional training resources. Unless otherwise stated, all experiments use the following configuration:

- **Backbone:** CLIP ViT-B/16 (frozen image encoder).

- **Training epochs per task:** 60.
- **Batch size:** 64.
- **Optimizer:** Adam.
- **Learning rate:** 0.001.
- **Random seed:** 1993.
- **Data usage:** No external data; each task uses only its designated subset.

LACE does not introduce any additional training epochs. The Concept Attribution (CA) and Concept Verification (CV) steps are implemented as lightweight operations on the concept activations of the current task:

- CA requires at most one additional pass over the task dataset $D_t$ to compute concept gradients.
- CV operates on the linear classifier in concept space and involves small-scale IRLS and hat-matrix computations.

Both components reuse the same training data and do not alter the number of forward/backward passes through the backbone. Thus, all methods are trained under an equivalent compute budget, ensuring a fair comparison.

### F.1 ADDITIONAL RESULTS ON TRAINING COST AND MEMORY OVERHEAD

In this section, we provide the empirical measurements that support our claim that LACE is a *lightweight* continual learning framework. As discussed in the main text, LACE adds only small concept-level heads and lightweight CA/CV computations on top of a frozen CLIP encoder. To quantify this, we report the end-to-end training time, per-task cost, and memory overhead on the CUB-200 dataset with ViT-B/16 under two commonly used class-incremental settings. All models use the same compute configuration (see Appendix F) and a pruning-rate bound of 0.1.

Table 3: Training cost and additional memory overhead introduced by LACE. "Extra memory" refers to the overhead beyond a frozen CLIP ViT-B/16 encoder and a standard linear classifier.

| Dataset | Model | Setting | Pruning rate | Time per last task (s) | Total time (s) | Total time (hours) | Extra memory |
|---------|-------|---------|--------------|------------------------|----------------|--------------------|--------------|
| CUB-200 | ViT-B/16 | B10 Inc10 | 0.1 | 55 | 12660 | 3h 31m | $\approx 1$ MB $\pm 0.1$ |
| CUB-200 | ViT-B/16 | B100 Inc10 | 0.1 | 56 | 5340 | 1h 29m | $\approx 0.9$ MB $\pm 0.1$ |

These results show that:

- The **additional memory overhead** from concept parameters, CA/CV buffers, and prototype augmentation is only around ~1 MB, which is negligible compared to the CLIP ViT-B/16 encoder.
- The **training time per incremental task** is small (55–56 seconds), demonstrating that CA and CV introduce only minimal computation.
- The **total cost for the full continual learning sequence** remains modest (1.5–3.5 hours), even though it spans 11–20 tasks depending on the split.

Collectively, these measurements justify describing LACE as *lightweight*: it retains scalability and efficiency while providing substantial performance and interpretability benefits.

## G ADDITIONAL ABLATIONS AND QUALITATIVE ANALYSIS

### G.1 ABLATION OF CONCEPT ATTRIBUTION (CA) AND CONCEPT VERIFICATION (CV)

To disentangle the contributions of Concept Attribution (CA) and Concept Verification (CV), we conduct an ablation study on **CUB-200** with **ViT-B/16** under the **B100 Inc10** setting. We consider three variants:

Table 4: Ablation of CA and CV on CUB-200 with ViT-B/16 (B100 Inc10).

| Dataset | Model | Setting | Components | $\bar{A}$ |
|---------|-------|---------|------------|-----------|
| CUB-200 | ViT-B/16 | B100 Inc10 | CA only | 81.53 |
| CUB-200 | ViT-B/16 | B100 Inc10 | CV only | 81.61 |
| CUB-200 | ViT-B/16 | B100 Inc10 | CA + CV (full LACE) | **82.64** |

- **CA only**: LACE with CA but without CV (we use a fixed pruning ratio instead of data-driven verification).
- **CV only**: LACE with CV but replacing CA with a naive magnitude-based ranking of concepts.
- **CA + CV (full LACE)**: our full method using both CA and CV.

The average accuracy $\bar{A}$ over all tasks is reported in Table 4.

Both CA and CV individually improve performance compared to a naive baseline, but the full LACE model (CA + CV) achieves the best result with $\bar{A} = 82.64\%$, which is about $+1$ percentage point higher than either CA-only or CV-only. This supports our design choice that CA and CV are complementary: CA provides principled concept importance scores (satisfying the *Sensitivity* and *Implementation Invariance* axioms), while CV uses a hat-matrix-based LOOCV approximation (Section 3.5) to automatically select an appropriate pruning ratio for each task.

### G.2    QUALITATIVE ANALYSIS OF PRUNED VS. RETAINED CONCEPTS

We now provide a qualitative analysis of which concepts are pruned versus retained by LACE. Figure 2 visualizes one example class from **CUB-200** under the **ViT-B/16** backbone.

In the top panel, we list concepts that are *pruned* by CV, together with representative images. These concepts predominantly describe background or contextual information, such as *"soar above the choppy, blue ocean waters nearby"*, *"pale pink"*, or *"sharp and slightly curved long, slender wings"*. Such descriptions tend to capture scene context or generic motion and are not consistently discriminative for the target class across tasks.

In the bottom panel, we show *retained* concepts. These focus on stable, class-specific attributes of the bird, e.g., *"glossy black feathers that shimmer in the sunlight"*, *"eyes are small and framed by white feathers"*, and *"stands confidently on sturdy legs"*. These retained concepts correspond to morphology and appearance features that are consistently useful for distinguishing the class.

Overall, this visualization illustrates that CV tends to remove concepts tied to transient backgrounds or loosely related descriptions, while preserving compact sets of semantically coherent, discriminative concepts. This supports our claim that LACE improves interpretability by maintaining a small yet meaningful concept bottleneck throughout continual learning.

## H    NOTATION TABLE

Table 5 summarizes the key mathematical symbols and notations used in the description of the LACE framework.

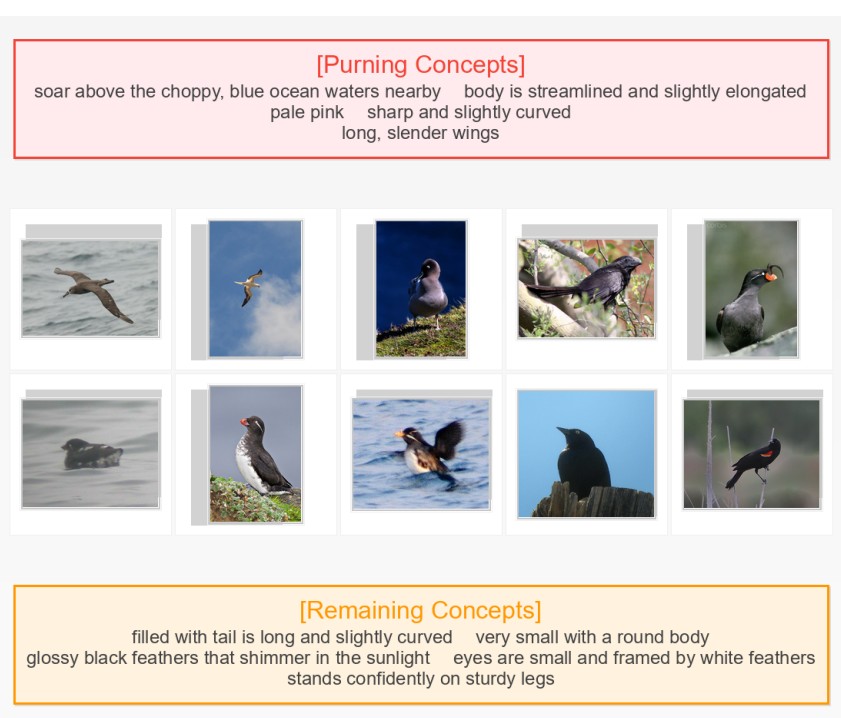

Figure 2: Qualitative visualization on CUB-200 with ViT-B/16. Top: concepts pruned by CV and corresponding images; many describe background or generic context. Bottom: concepts retained by LACE, which focus on stable, discriminative attributes of the bird (e.g., color, feather texture, and posture).

Table 5: Summary of Mathematical Notations.

| Symbol | Description |
|---|---|
| *Problem Definition & Data* | |
| $\mathcal{T}$ | Task index set $\{1, 2, \ldots, T\}$. |
| $\mathcal{D}_t$ | Dataset available for the current task $t$, consisting of pairs $(x_i, y_i)$. |
| $\hat{\mathcal{D}}_t$ | Final training set for task $t$, including augmented pseudo-features ($\mathcal{D}_t \cup \mathcal{D}_t^a$). |
| $\mathcal{Y}_t, \mathcal{Y}_{1:t}$ | Label set for task $t$ and cumulative label space up to task $t$. |
| $E_I(\cdot), E_T(\cdot)$ | CLIP image and text encoders, mapping to $\mathbb{R}^d$ ($d = 512$). |
| *CBM Architecture & Prototype Augmentation* | |
| $\mathcal{P}_t, \hat{\mathcal{P}}_t$ | Concept pool for task $t$ and cumulative concept pool $\bigcup_{\tau \leq t} \mathcal{P}_\tau$. |
| $M_t, \hat{M}_t$ | Number of generated concepts and number of retained concepts for task $t$. |
| $\mathbf{C}_t$ | Bank of concept vectors encoded by $E_T(\cdot)$ for task $t$. |
| $\mathbf{W}_c^t$ | Learnable parameters of the Concept Bottleneck Layer (CBL) at task $t$. |
| $\mathbf{W}_l^t$ | Learnable parameters of the linear classification head at task $t$. |
| $\mathcal{L}_{\text{sim}}$ | Similarity loss aligning concept scores with CLIP's concept space. |
| $P_i$ | Class prototype calculated as the mean image embedding for class $i$. |
| $\mathbf{V}_{\text{pseudo}}^j$ | Set of pseudo-feature vectors generated for old class $j$ using semantic matching. |
| *Concept Attribution (CA)* | |
| $E_i^g$ | Concept representation of sample $i$ at exploration step $g$. |
| $\Delta E_i^g$ | Increment of concept representation between steps $g$ and $g + 1$. |
| $\eta$ | Step size (learning rate) for the exploration update in concept space. |
| $A_j^t$ | Importance score of concept $j$ at task $t$ (discrete path-integral approximation). |
| *Concept Verification (CV)* | |
| $\text{CV}_k$ | Approximated Leave-One-Out Cross-Validation score for pruning ratio $k$. |
| $H_k$ | Hat matrix used to approximate the effective complexity of the model. |
| $\gamma_k$ | Effective complexity term, approximated by $\text{tr}(H_k)$. |
| $\tilde{X}_k$ | Scaled design matrix used in the IRLS approximation. |
| $S_i$ | Softmax covariance matrix for sample $i$, $S_i \in \mathbb{R}^{C \times C}$. |

