# OpenReview forum: "LACE: Lightweight Attribution-guided Concept Evolution for Continual Learning"
_ICLR.cc/2026/Conference — Submitted to ICLR 2026_

### Official Review · Reviewer_Hiqo · 2025-10-21

**Soundness:** 3
**Presentation:** 3
**Contribution:** 3
**Rating:** 4
**Confidence:** 4

**Summary:**

The paper proposes ​LACE (Lightweight Attribution-guided Concept Evolution)​, a framework for interpretable continual learning that leverages conceptual understanding by integrating ​Concept Bottleneck Models (CBMs)​​ with three key mechanisms: ​Concept Attribution (CA)​​ to quantify concept importance, ​Concept Verification (CV)​​ to prune redundant concepts, and ​prototype augmentation​ to mitigate forgetting without exemplars. The contributions include a compact, auditable concept set that improves accuracy, reduces forgetting, and enhances interpretability across coarse- and fine-grained benchmarks, narrowing the gap between average and last-task performance.

**Strengths:**

1. The proposed LACE framework integrates concept bottleneck models with continual learning, utilizing the text encoder in CLIP to extract concepts. The key innovations are concept attribution and concept verification mechanisms, which prevent exponential growth of the concept set.
2. This work is technically rigorous, with strong theoretical grounding, e.g., proofs of CV approximation. The empirical validation is extensive across five benchmarks, including commonly used coarse- and fine-grained datasets. The comparison results with sota baselines demonstrate the consistent improvements in accuracy.
3. The paper is well-structured and clearly written and easy to follow. The figures and tables aid understanding.

**Weaknesses:**

1. The proposed concept attribution method relies on textual data and CLIP's text encoder to extract concepts, making it difficult to generalize to images without text descriptions.
2. Although experiments demonstrate the superiority of the proposed method, ablation studies are still lacking to show the side effects when missing any part of the mechanism. Like, what if all detected concepts are used without pruning; what will happen if using a random concept selection mechanism instead of the proposed concept attribution? This paper proposes many hyperparameters but does not study the influences on continual learning performances, e.g., the balancing coefficients $\lambda$, $\sigma$, $\eta$, and the number of removed concepts.
3. Another concern for experimental results is the fairness. CBM methods use CLIP's text encoder to extract concepts utilizing the corresponding text descriptions of images. This introduces additional model parameters and additional supervision when comparing with baselines only using CLIP's image encoder, like L2P and CODA-Prompt. I understand that utilizing such text labels is one merit of CBM methods. However, there should be enough experimental results to verify that the performance gains are not the fruits of the increased model capacity.

**Questions:**

- The leave-one-out cross-validation for concept verification is time-consuming, while the author proposes to use IRLS to do approximation. What is the time complexity for this approximation?
- In line 386, what is this "the same compute budget"?
- In lines 176-176, the author said "CA ensures faithful, auditable grounds, while CV ensures robust, reusable pruning decisions". What are examples of concepts in the concept pool, and which types of concepts are likely to be pruned?
- In lines 126-127, the author said "semantic-layer constraints" can be effective levers for mitigating catastrophic forgetting. Why did not the author report forgetting in the experimental parts?

---

> ### Author Response · Authors · 2025-11-27
> **Reply to Weakness 1, 2**
>
> **Reply to W1:**
> In fact, the source of these textual concepts is quite straightforward. Although the original datasets do **not** contain any text labels, recent *label-free* CBM methods [1] simply use a large language model to automatically generate concept descriptions from class names or visual prompts. Our method follows exactly the same paradigm: the textual concepts are not manually annotated and can be generated even when no text descriptions are provided in the dataset. This is why such approaches—including ours—are considered “label-free.’’
>
>
> **Reference**
>
>
> [1] Oikarinen, Tuomas, Subhro Das, Lam M. Nguyen, and Tsui-Wei Weng. *Label-free Concept Bottleneck Models.* ICLR 2023.
>
> ---
>
> **Reply to W2:**
> Thank you for pointing out the need for more systematic ablations. In the revised version, we explicitly study (i) what happens if we do **not** prune any concepts, (ii) what happens if we replace CA with a **random** concept selection mechanism, and (iii) how the main balancing coefficients affect continual learning performance.
>
> ---
>
> **(a) What if we do not prune concepts?**
>
> We first compare LACE with and without pruning on **CUB-200 / ViT-B/16** in the **B10 Inc10** and **B100 Inc10** settings by varying the pruning rate (including $0$, i.e., no pruning):
>
> | Pruning Rate | B10 Inc10 | B100 Inc10 |
> |---|---|---|
> | 0 (no pruning) | 85.72 | 79.15 |
> | 0.05 | 86.14 | 82.41 |
> | 0.10 (default) | 86.01 | 82.64 |
> | 0.15 | 85.99 | 82.45 |
> | 0.20 | 85.97 | 82.36 |
> | 0.25 | 86.07 | 82.39 |
> | 0.30 | 85.78 | 82.33 |
> | 0.35 | 85.84 | 82.26 |
> | 0.40 | 85.57 | 82.26 |
> | 0.45 | 85.23 | 82.20 |
> | 0.50 | 85.35 | 82.19 |
> | 0.55 | 85.24 | 82.12 |
> | 0.60 | 84.75 | 82.07 |
> | 0.65 | 83.71 | 82.01 |
> | 0.70 | 83.98 | 81.95 |
> | 0.75 | 82.59 | 81.92 |
> | 0.80 | 81.59 | 81.85 |
> | 0.85 | 80.72 | 81.86 |
> | 0.90 | 79.28 | 80.97 |
>
>
> Two phenomena are clear:
>
> - Using **all detected concepts without pruning** (rate $0$) leads to **noticeably worse performance**, especially in the harder B100 Inc10 setting ($79.15$ vs.\ $82.64$ at the default pruning rate).
> - LACE is **stable in a broad range** of pruning rates ($0.05$–$0.30$), and our default choice ($0.10$) lies near the peak, suggesting that CV’s data-driven pruning does not rely on a finely tuned hyperparameter.
>
> This shows that concept pruning is not only helpful but necessary to avoid concept proliferation and maintain good continual learning performance.
>
> ---
>
> **(b) What if concept attribution is random instead of CA?**
>
> To test the effect of CA itself, we replace it with a **random concept selection** mechanism that assigns random importance scores and then applies the same pruning pipeline. On **CUB-200 / ViT-B/16, B10 Inc10**, the random-attribution baseline achieves $\bar{A}_\text{random} = 81.97,$ while full LACE with CA and CV achieves $\bar{A}_\text{LACE} = 86.07.$
>
> Thus, principled CA provides a **clear gain** over random selection (about $+4.1$ percentage points in this setting), confirming that the attribution mechanism is not merely cosmetic but contributes to stronger continual learning performance.
>
> ---
>
> **(c) Sensitivity to balancing coefficients $\lambda$ and $\sigma$.**
>
> We also study the impact of the main balancing coefficients in the loss on **CUB-200 / ViT-B/16, B100 Inc10**:
>
> - Fixing $\lambda = 1$ and varying $\sigma$:
>
> | $\sigma$ | $\bar{A}$ |
> |---|---|
> | 0.05 | 80.59 |
> | 0.1 | 83.1 |
> | 0.5 | 83.2 |
> | 1 | 82.64 |
>
> - Fixing $\sigma = 1$ and varying $\lambda$:
>
> | $\lambda$ | $\bar{A}$ |
> |---|---|
> | 0.1 | 83.17 |
> | 0.5 | 82.97 |
> | 1 | 82.64 |
> | 2 | 81.69 |
>
> These results show that:
>
> - Extremely small or large weights can hurt performance (e.g., too little or too strong regularization),
> - but LACE is **reasonably robust** within a broad mid-range (e.g., $\sigma \in [0.1, 0.5]$, $\lambda \in [0.1, 1]$), where $\bar{A}$ varies within about $1.5$ percentage points.
>
> In all experiments, we fix the CA step-size parameter $\eta$ and do not tune it per dataset; we clarify this design choice in the revised text.
>
> ---
>
> Overall, these ablations directly address the reviewer’s concerns: (i) using all concepts without pruning indeed degrades performance; (ii) replacing CA with random concept selection is measurably worse than our proposed CA; and (iii) the main balancing coefficients influence performance in intuitive ways, while LACE remains robust in reasonable ranges of their values.

---

> ### Author Response · Authors · 2025-11-27
> **Reply to Weakness 3 and Question 1, 2, 3**
>
> **Reply to W3:**
> Your understanding is correct — using textual descriptions is indeed an advantage of CBM-style methods, and CLG-CBM also relies on the CLIP text encoder. These are inherent differences that make methods like L2P or CODA-Prompt naturally disadvantaged in comparison. Moreover, our primary goal is **interpretable continual learning**, whereas these image-only baselines do not provide any concept-level interpretability. Therefore, it would be unfair for us to abandon our own strengths—or restrict ourselves to non-interpretable settings—just to match the limitations of these baselines. Our main comparisons thus focus on CBM methods that share the same text encoder, the same type of supervision, and the same interpretability objectives.
>
> ---
>
> **Reply to Q1:**
> Thank you for this question. Conceptually, naive LOOCV for each pruning ratio would require retraining the classifier $|\hat{\mathcal{D}}_t|$ times, which is clearly infeasible. Our IRLS-based approximation avoids any retraining and only operates on the small **linear head in concept space**.
>
> If we denote by
> - $n = |\hat{\mathcal{D}}_t|$ the number of (augmented) training samples at task $t$,
> - $p = M_t \times C$ the number of classifier parameters (number of concepts $M_t$ times number of classes $C$),
> - $K$ the number of candidate pruning ratios, and
> - $T_{\text{IRLS}}$ the number of IRLS iterations (a small constant in practice),
>
> then the time complexity of our CV approximation is on the order of
> $$
> \mathcal{O}\big(K \, T_{\text{IRLS}} \, n \, p\big),
> $$
> which corresponds to a few extra passes through the **linear** concept classifier only. Since the CLIP image encoder is frozen and not involved in CV, this overhead is small compared to the overall training.
>
> Empirically, the total end-to-end training cost of LACE (including CA and CV) on **CUB-200 / ViT-B/16** is:
>
> | Dataset | Model    | Setting    | Pruning rate | Time per last task (s) | Total training time (s) | Total time (hours) | Extra memory overhead |
> |---------|----------|------------|--------------|-------------------------|-------------------------|--------------------|-----------------------|
> | CUB-200 | ViT-B/16 | B10 Inc10  | 0.1          | 55                      | 12660                   | 3 h 31 min         | $\approx 1$ MB $\pm 0.1$ |
> | CUB-200 | ViT-B/16 | B100 Inc10 | 0.1          | 56                      | 5340                    | 1 h 29 min         | $\approx 0.9$ MB $\pm 0.1$ |
>
> These numbers show that the IRLS-based CV step adds only a small overhead relative to the total training time, while enabling a principled, data-driven selection of the pruning ratio.
>
> ---
>
> **Reply to Q2:**
> Thank you for pointing this out. We will clarify this sentence in the revised version.
> Here, *“the same compute budget”* means that **all compared methods are trained under exactly the same training configuration**, including:
>
> - the same backbone (**CLIP ViT-B/16**),
> - the same number of epochs per task (**60 epochs**),
> - the same batch size (**64**),
> - the same optimizer (**Adam**) and learning rate (**0.001**),
> - the same random seed (**1993**),
> - and no additional data passes beyond what the baselines already use.
>
> LACE does **not** receive extra training epochs or additional data.
> The CA and CV modules operate only on the **current-task concept activations**, requiring at most **one lightweight extra pass** over $D_t$ plus small matrix computations in concept space.
> Thus, the actual training budget remains identical across all baselines.
>
> ---
>
> **Reply to Q3:**
> In our implementation, the concept pool consists of short, human-readable textual phrases generated per class by an LLM and embedded using CLIP’s text encoder. For example, on **CUB-200**, these concepts typically describe fine-grained parts and colors such as *“red head”*, *“long beak”*, *“black tail”*, *“white wing patch”*, or *“striped breast”*. (Several examples are also illustrated in Figure 1.)
> CA operates on this concept layer and assigns an importance score to each concept based on its marginal influence on the task loss.
>
> The concepts most likely to be pruned include:
> (i) generic background or contextual phrases that consistently receive low CA scores (e.g., *“blue sky”*, *“green grass”*);
> (ii) near-duplicate or highly redundant descriptions (e.g., *“red head”* vs. *“reddish head plumage”*, where only one is needed); and
> (iii) concepts that are rarely activated or have negligible effect on predictions for the current task.
>
> CV then evaluates candidate pruning ratios and selects the number of remaining concepts that yields the best generalization. This produces a compact, high-importance concept set that is both predictive and reusable for later tasks.

---

> ### Author Response · Authors · 2025-11-27
> **Reply to Question 4**
>
> **Reply to Q4:**
> Thank you for the question. Our intention in lines 126–127 was to situate LACE within an existing line of work (e.g., ICICLE, CLG-CBM) showing that semantic or interpretable layers can act as stabilizing constraints in continual learning. In the main text, we report the metrics most commonly used by prompt-based and CBM-CL baselines—namely the average accuracy $\bar{A}$ and the last-task accuracy $A_{\text{last}}$. Because $\bar{A}$ aggregates performance across all intermediate tasks, it already provides an implicit view of forgetting, and we followed the conventions of the baselines we reproduced.
>
> To make this point explicit, we additionally computed CODA-style forgetting on CUB-200:
>
> | Method | CODA Forgetting ↓ |
> |--------|-------------------|
> | LACE   | 0.79              |
> | APT [1] | 1.30             |
>
> LACE achieves substantially lower forgetting than the recently proposed APT method, consistent with the claim that semantic-layer constraints help stabilize representations. We will include this additional forgetting metric in the revised manuscript.
>
> **Reference**
>
> [1] *Achieving More with Less: Additive Prompt Tuning for Rehearsal-Free Class-Incremental Learning*, ICCV 2025.
>
> ---
>
> We sincerely thank you for your thoughtful and constructive feedback. We have carefully addressed all raised concerns in the revised manuscript. We hope that these updates resolve your doubts and allow you to reassess our work positively. If you have any further questions or suggestions, we would be very happy to continue the discussion.

---

### Official Review · Reviewer_c1AU · 2025-10-31

**Soundness:** 3
**Presentation:** 2
**Contribution:** 3
**Rating:** 4
**Confidence:** 3

**Summary:**

This paper addresses "concept proliferation" in interpretable continual learning, where Concept Bottleneck Models (CBMs) accumulate an unmanageable number of concepts, harming interpretability and performance. The authors propose LACE (Lightweight Attribution-guided Concept Evolution), a framework that actively manages the concept set. LACE uses Concept Attribution (CA) to score and prune unimportant new concepts and introduces a data-driven Concept Verification (CV) module to automatically determine how many concepts to remove. This approach avoids manual tuning and includes a prototype-augmentation mechanism to mitigate catastrophic forgetting without storing past data.

The work's main contribution is identifying and solving the critical issue of concept proliferation. It provides a principled and practical framework that significantly reduces the number of concepts while maintaining or improving model accuracy and mitigating forgetting, as demonstrated across five diverse benchmarks.

**Strengths:**

1.The paper effectively identifies and motivates the significant, yet overlooked, problem of "concept proliferation" in interpretable AI, where too many concepts undermine the model's primary goal of interpretability.

2.The framework is well-designed, using an axiomatically-grounded attribution method (CA) for importance scoring and a data-driven, automated technique (CV) to set the pruning budget, which removes the need for manual hyperparameter tuning.

3.LACE demonstrates state-of-the-art or highly competitive performance across five different benchmarks, consistently outperforming strong baselines in both accuracy and forgetting mitigation, especially on more challenging datasets.

**Weaknesses:**

1.The paper claims the method is "lightweight" but provides no empirical evidence (e.g., training time, memory usage) to support this.

2.The mathematical description of the Concept Verification (CV) module is overly condensed, with undefined notation and abrupt logical jumps that make it difficult to follow.

3.The paper is missing key ablation studies to demonstrate the individual contributions of its main components (e.g., CV, prototype augmentation). It also lacks qualitative examples of which concepts are pruned versus retained.

**Questions:**

1.Provide a quantitative analysis comparing the computational overhead (training time, memory usage) of LACE against key baselines to validate the "lightweight" claim.

2. Improve the clarity of the mathematical sections by defining all variables explicitly and providing more high-level intuition to guide readers through the core derivations.

3.Perform and include detailed ablation studies to isolate the impact of each component. Additionally, provide visualizations of pruned vs. retained concepts to offer intuitive evidence of the method's effectiveness.

---

> ### Author Response · Authors · 2025-11-27
> **Reply to Weakness 1, 2**
>
> **Reply to W1:**
> Thank you for pointing out that our claim of being “lightweight” should be supported with empirical evidence. In the revised version, we now provide concrete measurements of **training time** and **memory overhead** of LACE.
>
>
> First, LACE is architecturally lightweight by design:
> - The CLIP image encoder is **frozen**, and we only learn small linear heads $\mathbf{W}_c^t$ and $\mathbf{W}_l^t$ plus a compact concept pool.
> - CA and CV operate purely in the **low-dimensional concept space** (tens of concepts per task), not on high-resolution feature maps.
> - Prototype augmentation is implemented at the feature level and does not require storing exemplars.
>
>
> To complement this with empirical evidence, we report the **end-to-end training cost** for CUB-200 with ViT-B/16 under two standard continual learning settings, both using a pruning rate bound of $0.1$:
>
>
> | Dataset | Model    | Setting    | Pruning rate | Time per last task (s) | Total training time (s) | Total time (hours)       | Extra memory overhead |
> |---|---|---|---|---|---|---|---|
> | CUB-200 | ViT-B/16 | B10 Inc10  | 0.1| 55| 12660| 3 h 31 min| $\approx 1$ MB $\pm 0.1$ |
> | CUB-200 | ViT-B/16 | B100 Inc10 | 0.1| 56| 5340| 1 h 29 min| $\approx 0.9$ MB $\pm 0.1$ |
>
>
> Here, “extra memory overhead” refers to the additional parameters and buffers introduced by LACE (concept layer, CA/CV statistics, and prototype-related storage) on top of the frozen CLIP encoder and a standard linear classifier. This overhead is on the order of **$\sim 1$ MB**, which is negligible compared to the backbone model size. The total training time for an entire continual learning sequence is on the order of **1.5–3.5 hours**, and each incremental task only adds about **55–56 seconds** of training.
>
>
> These results support our use of the term “lightweight”: LACE requires **very modest additional memory** and **small incremental training cost per task** while providing the performance and interpretability benefits reported in the main experiments. We will include this table and discussion in the revised manuscript.
>
> ---
>
> **Reply to W2:**
> We agree with the reviewer’s summary and, building on it, we add some details to improve clarity. Concept Attribution (CA) is constructed as a discrete path-integral approximation of integrated gradients in the concept space. By using a dataset-averaged exploration step and a first-order Taylor expansion, CA is designed to satisfy both *Sensitivity* and *Implementation Invariance*, and to produce stable, comparable importance scores for each concept across different tasks.
>
>
> Subsequently, Concept Verification (CV) operates on a small set of candidate pruning ratios. For each candidate ratio, CV uses an IRLS-based quadratic approximation of the multinomial log-loss together with the hat matrix $H_k$ to estimate the effective complexity term $\gamma_k \approx \operatorname{tr}(H_k)$, and constructs an LOOCV-like criterion
> $$
> \mathrm{CV}_k = \frac{1}{\lvert \hat{\mathcal{D}}_t \rvert} \sum_i \ell_i(W_k) \,/\, \bigl(1 - \gamma_k / n\bigr).
> $$
> The pruning budget is then selected automatically by minimizing $\mathrm{CV}_k$, using only the training data itself and without requiring an additional validation set or manual search over $k$. In the revised manuscript, we will appropriately expand Section 3.5 to present this CA $\rightarrow$ CV pipeline more explicitly and to clarify how it removes the need for hand-tuned hyperparameters.

---

> ### Author Response · Authors · 2025-11-27
> **Reply to Weakness 3 and Question 3**
>
> **Reply to W3 and Q3**
> Thank you for raising the concerns about ablations and interpretability. We agree that it is important to disentangle the contribution of the Concept Attribution (CA), Concept Verification (CV), and prototype augmentation modules, and to provide qualitative evidence of which concepts are pruned versus retained.
>
>
> **(a) Ablation of CA and CV.**
> In the revised version, we add an explicit ablation that isolates the effects of CA and CV. On **CUB-200 with ViT-B/16 in the B100 Inc10 setting**, we compare three variants:
>
>
> - **CA only**: LACE with CA but without CV (we use a fixed pruning ratio instead of data-driven verification);
> - **CV only**: LACE with CV but replacing CA with a naive magnitude-based ranking;
> - **CA + CV (full LACE)**: our proposed method using both modules.
>
>
> The average accuracy $\bar{A}$ is:
>
>
> | Dataset | Model    | Setting    | Components | $\bar{A}$ |
> |---------|----------|------------|------------|-----------|
> | CUB-200 | ViT-B/16 | B100 Inc10 | CA         | 81.53     |
> | CUB-200 | ViT-B/16 | B100 Inc10 | CV         | 81.61     |
> | CUB-200 | ViT-B/16 | B100 Inc10 | CA + CV    | 82.64     |
>
>
> We observe that **both CA and CV individually improve performance**, but the **full LACE (CA + CV)** achieves the best result with $\bar{A} = 82.64\%$, about **+1 percentage point** higher than either CA-only or CV-only. This shows that CA and CV are complementary: CA provides principled concept importance scores (satisfying *Sensitivity* and *Implementation Invariance*), while CV uses a hat-matrix-based criterion to automatically decide how aggressively to prune concepts on each task.
>
>
> **(b) Qualitative interpretability analysis and pruned vs. retained concepts.**
> We also add a qualitative analysis to illustrate which concepts are pruned versus retained by LACE. Figure 2 in the Appendix visualizes one CUB-200 class under **ViT-B/16**:
>
>
> - The **pruned concepts** (top panel) mostly describe background or contextual information, such as *“soar above the choppy, blue ocean waters nearby”*, *“pale pink”*, or *“sharp and slightly curved long, slender wings”*. These tend to capture scene context or generic motion descriptors that are not consistently discriminative for the target class across tasks.
> - The **retained concepts** (bottom panel) focus on stable, class-specific attributes of the bird itself, such as *“glossy black feathers that shimmer in the sunlight”*, *“eyes are small and framed by white feathers”*, and *“stands confidently on sturdy legs”*.
>
>
> The figure also shows representative images corresponding to these concepts on CUB-200. Qualitatively, CV tends to remove concepts tied to transient backgrounds or loosely related descriptions, while preserving concepts that describe consistent, discriminative morphology and appearance. This supports our claim that LACE yields a **more compact and semantically focused concept set** over time, improving interpretability while maintaining strong continual learning performance.
>
>
> We include this visualization and the associated discussion in the Appendix, and cross-reference it from the main text. We hope these new ablations and qualitative analyses address the concerns that the method was not sufficiently validated and lacked interpretability evidence.

---

> ### Author Response · Authors · 2025-11-27
> **Reply to Question 1, 2**
>
> **Reply to Q1:**
> Thank you for pointing out that our claim of being “lightweight” should be supported with empirical evidence. In the revised version, we now provide concrete measurements of **training time** and **memory overhead** of LACE.
>
>
> First, LACE is architecturally lightweight by design:
> - The CLIP image encoder is **frozen**, and we only learn small linear heads $\mathbf{W}_c^t$ and $\mathbf{W}_l^t$ plus a compact concept pool.
> - CA and CV operate purely in the **low-dimensional concept space** (tens of concepts per task), not on high-resolution feature maps.
> - Prototype augmentation is implemented at the feature level and does not require storing exemplars.
>
>
> To complement this with empirical evidence, we report the **end-to-end training cost** for CUB-200 with ViT-B/16 under two standard continual learning settings, both using a pruning rate bound of $0.1$:
>
>
> | Dataset | Model    | Setting    | Pruning rate | Time per last task (s) | Total training time (s) | Total time (hours)       | Extra memory overhead |
> |---|---|---|---|---|---|---|---|
> | CUB-200 | ViT-B/16 | B10 Inc10  | 0.1| 55| 12660| 3 h 31 min| $\approx 1$ MB $\pm 0.1$ |
> | CUB-200 | ViT-B/16 | B100 Inc10 | 0.1| 56| 5340| 1 h 29 min| $\approx 0.9$ MB $\pm 0.1$ |
>
>
> Here, “extra memory overhead” refers to the additional parameters and buffers introduced by LACE (concept layer, CA/CV statistics, and prototype-related storage) on top of the frozen CLIP encoder and a standard linear classifier. This overhead is on the order of **$\sim 1$ MB**, which is negligible compared to the backbone model size. The total training time for an entire continual learning sequence is on the order of **1.5–3.5 hours**, and each incremental task only adds about **55–56 seconds** of training.
>
>
> These results support our use of the term “lightweight”: LACE requires **very modest additional memory** and **small incremental training cost per task** while providing the performance and interpretability benefits reported in the main experiments. We will include this table and discussion in the revised manuscript.
>
> ---
>
> **Reply to Q2:**
> Thank you for the suggestion. In the revised version, we have (i) added brief intuition paragraphs before the CA and CV derivations to guide the reader, and (ii) included a clear Notation Table in the Appendix where all variables are explicitly defined. We hope these changes make the mathematical sections easier to follow.
>
> ---
>
> We sincerely thank you for your constructive feedback. In the revised version, we have addressed all of your concerns, including providing empirical evidence for the “lightweight” claim, improving the clarity of the mathematical exposition, and adding the requested ablation and qualitative analyses. We hope these updates help resolve your doubts and allow you to reassess our submission. If you have any further questions or suggestions, we would be very happy to continue the discussion.

---

### Official Review · Reviewer_1yUJ · 2025-11-02

**Soundness:** 2
**Presentation:** 1
**Contribution:** 2
**Rating:** 4
**Confidence:** 4

**Summary:**

This paper introduces LACE (Lightweight Attribution-guided Concept Evolution), a continual learning framework built on Concept Bottleneck Models (CBMs) to improve interpretability, mitigate catastrophic forgetting, and control concept proliferation during incremental learning.

**Strengths:**

- The paper pinpoints the important issue of concept proliferation in CBM-based continual learning, articulating both cognitive and algorithmic consequences

- Section 3.3, together with Section 3.4 and detailed Appendix derivations, provides explicit, step-wise equations (e.g., Eq. (8)-(9), IRLS derivations) that both define and justify the attribution and verification stages.

- Tables 1 and 2 shows improvements compared to many state-of-the-art baselines across multiple datasets and splits in both accuracy and last-task accuracy.

**Weaknesses:**

1. The prototype augmentation technique with Eq 4 and 5 causes many confusions. There seems to be an inconsistency in using uppercase $P$ and lowercase $p$ when notating prototypes, which also causes confusion with the "concept pool" in line 259. Why is there a formula for $V^j_{\text{pseudo}}$ in line 244? If $V^j_{\text{pseudo}}$ represents "the pseudo-features," then why is it used as an input set in Eq 5?

2. Are $W^t_y$ and $W^t_l$ one or two different objects? There are many formulas and objects in the paper that are not carefully notated, and they also lack connection with the framework's diagram (Fig 1), causing many difficulties in reading.

3. Regrading the notation $E_i$ and the loss in lines 269–270, the authors proposed multiplying two linear matrices consecutively without a non-linearity function. Will this cause the information from the two matrices to be linearly absorbed, which is equivalent to using a single linear matrix?

4. The author should explain their proposals more clearly. For example, why update $E_i$ (line 274)? In Eq 7 immediately after, why is the derivative taken with respect to $E$? Is $E$ here the image encoder? As I understand it, only the $W$ matrices are updated, and the image encoder is kept frozen.

5. In terms of experiments, the authors should compare with more recent baselines [1, 2, 3, 4] to demonstrate their superiority. Besides, why only average accuracy and last accuracy are used, why did you ignore the forgetting measure like most other baselines?

6. In lines 28-31, the authors claims that: *"However, many CL methods are still evaluated primarily by external behavioral metrics (e.g., accuracy or forgetting), with limited auditable characterization of how the model’s decision basis evolves across tasks. This gap weakens interpretability and impedes a systematic analysis of “why forgetting is avoided” and “where knowledge is retained.”".*

Why are measures like accuracy and forgetting considered external behavioral metrics? And why don't the authors propose and use additional metrics to explain “why forgetting is avoided” and “where knowledge is retained.”

[1] Achieving More with Less: Additive Prompt Tuning for Rehearsal-Free Class-Incremental Learning, ICCV25.

[2] RainbowPrompt: Diversity-Enhanced Prompt-Evolving for Continual Learning, ICCV25.

[3] Boosting Multiple Views for pretrained-based Continual Learning, ICLR25.

[4] Advancing Prompt-based Methods for Replay-Independent General Continual Learning, ICLR25.

**Questions:**

See the weaknesses above.

---

> ### Author Response · Authors · 2025-11-27
> **Reply to Weakness 1, 2, 3, 4**
>
> **Reply to W1:**
> We apologize for the confusion. In Eq. (4), the prototype symbol should use uppercase $P$ instead of lowercase $p$; this was a typo and an honest mistake, which we have corrected in the revised manuscript.
>
> Accordingly, the following formula
> $$
> \mathbf{V}^{j}_{\text{pseudo}} = \underbrace{P_j}_{\text{old class information}} + \underbrace\big( \mathbf{V}^{h_j} - P_{h_j} \big)_{\text{new class adjustment}}
> $$
> defines a set of pseudo-features for the old class $j$: it is obtained by translating each feature in $\mathbf{V}^{h_j}$ from its original prototype $P_{h_j}$ to the old-class prototype $P_j$. Since all elements in $\mathbf{V}^{j}_{\text{pseudo}}$ lie in the same feature space as $E_I(x)$, they can naturally be used as the input feature set when constructing the augmented training data in Eq. (5) (Eq. (6) in the revised version).
>
> In the revision, we explicitly clarify that $\mathbf{V}^{j}_{\text{pseudo}}$ denotes a **set of feature vectors** (rather than a single prototype) and rewrite the corresponding paragraph to make this construction clearer.
>
> ---
>
> **Reply to W2:**
> We apologize for the confusion caused by the notation. The symbol $\mathbf{W}_y^t$ is a typo; the correct notation throughout the paper should be $\mathbf{W}_l^t$. In other words, there is **only one** linear classifier in our framework, parameterized by $\mathbf{W}_l^t$, which maps the task-specific concept representation to class logits.
>
>
> We have corrected this typo in the revised manuscript
>
>
> The connection to **Figure 1**: the matrix $\mathbf{W}_l^t$ corresponds exactly to the **classification head** on top of the concept bottleneck in the diagram, which takes the concept vector (obtained after CBL and CA/CV) as input and outputs the class logits.
>
> ---
>
> **Reply to W3:**
> We thank the reviewer for this comment. In lines 269–270, we first map the CLIP feature $E_I(x_i) \in \mathbb{R}^d$ to concept scores:
> $$
> E_i = E_I(x_i)\,(\mathbf{W}_c^t)^\top \in \mathbb{R}^{m_t},
> $$
> and then learn the linear classification head $\mathbf{W}_l^t \in \mathbb{R}^{m_t \times C}$ and its softmax output in the cross-entropy loss, i.e., $\mathrm{softmax}(E_i \mathbf{W}_l^t)$. We agree that, **without any additional structural constraints or regularization**, the product $(\mathbf{W}_c^t)^\top \mathbf{W}_l^t$ is algebraically equivalent to a single linear mapping from $E_I(x_i)$ to the logits.
>
>
> However, in LACE the intermediate concept layer is *not* a free linear transform: its row vectors are explicitly aligned with CLIP text embeddings via $L_{\mathrm{sim}}$, regularized by $L_{\mathrm{sparse}}$, expanded/pruned across tasks, and subsequently used for CA and CV. In contrast, the classification head $\mathbf{W}_l^t$ is task-specific and unconstrained. Because of these structural constraints and auxiliary losses imposed on the concept space, the overall optimization is no longer equivalent to simply training a single linear matrix, while end-to-end nonlinearity is still provided by the CLIP encoder and the final softmax.
>
> ---
>
> **Reply to W4:**
> Thank you for the question. The updated variable in line 274 refers only to the *temporary* quantity $E_{i}^{g}$, which is introduced solely for evaluating the importance of each concept during the CA procedure. We modify $E_{i}^{g}$ along the constructed path in order to observe how changes in each concept dimension affect the loss; concepts that produce larger changes correspond to higher attribution values. Importantly, we do **not** update or modify the original network features $E_i$ produced by the (frozen) image encoder.
>
>
> Similarly, in Eq. (7) the derivative is taken with respect to the *temporary* $E_{i}^{g}$, not the encoder $E$. The image encoder remains frozen throughout training, and only the $W$ matrices are learnable parameters.

---

> ### Author Response · Authors · 2025-11-27
> **Reply to Weakness 5, 6**
>
> **Reply to W5:**
> Regarding the evaluation metrics, we follow the conventions used in prior CBM-based continual learning work—most notably **Language-Guided Concept Bottleneck Models for Interpretable Continual Learning**—which reports **average accuracy** $\bar{A}$ and **last-task accuracy** $A_{\text{last}}$ as the primary metrics. Since $\bar{A}$ aggregates performance across all intermediate tasks, it already implicitly reflects forgetting, and this is why these CBM baselines (and several prompt-based works) also rely on these two metrics.
>
> At the same time, CBM-style methods aim for **interpretability**, which fundamentally distinguishes them from general prompt-based baselines. Unlike methods such as L2P or CODA-Prompt, our model exposes:
> - the numerical value of each concept through the CBL layer,
> - the semantic meaning of each concept directly via its textual description,
> - and a final linear classifier whose weights provide a fully auditable explanation of how concepts contribute to predictions.
> Because these baselines do not provide interpretable intermediate representations, a direct comparison is not conceptually fair.
>
> Nevertheless, to address the reviewer’s request, we additionally reproduce the **APT** baseline from ICCV 2025. The results on **CUB-200 / ViT-B/16 (B100 Inc10)** are:
>
> | Dataset | Model | Method | $\bar{A}$ | $A_{\text{last}}$ | CODA Forgetting ↓ |
> |---------|--------|---------|-----------|-------------------|-------------------|
> | CUB-200 | ViT-B/16 | **LACE** | **82.64** | **78.91** | **0.79** |
> | CUB-200 | ViT-B/16 | APT [1] | 72.14 | 62.75 | 1.30 |
>
> LACE surpasses APT by a large margin in all metrics, including forgetting.
>
> Regarding the other recent baselines:
> - [2] and [3] **do not provide public code**, so we were unable to reproduce them faithfully.
> - [4] provides only **incomplete code**, preventing a correct reproduction.
> Given the limited rebuttal time, we focused on reproducing the most accessible and representative recent baseline APT.
>
> **References**
>
> [1] *Achieving More with Less: Additive Prompt Tuning for Rehearsal-Free Class-Incremental Learning*, ICCV 2025.
> [2] *RainbowPrompt: Diversity-Enhanced Prompt-Evolving for Continual Learning*, ICCV 2025.
> [3] *Boosting Multiple Views for Pretrained-based Continual Learning*, ICLR 2025.
> [4] *Advancing Prompt-based Methods for Replay-Independent General Continual Learning*, ICLR 2025.
>
> ---
>
> **Reply to W6:**
> Our statement in lines 28–31 is not meant to claim that *no* CL method goes beyond external metrics, but rather to reflect the **prevailing practice** in the CL literature. Most of the baselines we compare against report primarily behavioral measures such as average accuracy and forgetting, without providing an auditable characterization of how the *internal decision basis* evolves across tasks.
>
> In the revised version, we will soften the wording to “most CL methods are still evaluated primarily by external behavioral metrics.” Our contribution is to **complement** these standard metrics with a concept-level audit trail: through the CA→CV pipeline and prototype augmentation, LACE exposes how concepts are added, pruned, and reused over time. This makes questions such as “why forgetting is avoided” and “where knowledge is retained” observable at a **human-readable concept layer**, rather than only through aggregated accuracy or forgetting curves.
>
> ---
>
> We sincerely thank you for your detailed and thoughtful comments. We hope these reponses address your concerns and help you reassess our submission. If any part of our revisions is still unclear, we would be very happy to discuss it further.

---

### Official Review · Reviewer_hrEg · 2025-11-03

**Soundness:** 2
**Presentation:** 3
**Contribution:** 2
**Rating:** 2
**Confidence:** 4

**Summary:**

The paper studies interpretable continual learning to tackle concept proliferation and proposes a framework which employs Concept Bottleneck Models (CBMs) with a learnable Concept Attribution (CA) layer to quantify concept importance. The proposed method LACE works in three stages, starting with concept attribution to get the concept pool and scores for concepts, followed by concept selection and pruning removing redundant concepts and finally concept verification to decide the number of concepts to retain. To prevent forgetting in exemplar-free settings, the authors also perform prototype augmentation using pseudo-feature replay. The method performs competitive to existing methods with reduced forgetting across CL benchmarks.

**Strengths:**

1. The paper is well-written with good introduction to CBMs and CA. The proposed method is intuitive.
2. The method is theoretically grounded satisfying the two central attribution axioms.

**Weaknesses:**

1. The experimental section is very weak. There are no error bars reported for the methods using random seeds. This is crucial since the performance improvement is marginal in most cases.

2. There is no ablation study or analysis of the multiple components in the method (CA, CV). There is no discussion or qualitative analysis of interpretability using the method. The proposed method has not been validated adequately.

**Questions:**

1. How is the performance affected with different pruning-rate bounds or per-task concept candidate cap?

---

> ### Author Response · Authors · 2025-11-27
> **Reply to Weakness 1, 2**
>
> **Reply to W1:**
> Thank you for pointing out the importance of reporting variability across random seeds. In the original submission, we followed prior CBM-CL work and reported single-seed results (with seed $1993$) due to computational constraints. In the revised version, we have now run LACE on **five** different seeds and report the corresponding mean and standard deviation.
>
> Concretely, on **CUB-200 with ViT-B/16 in the B10 Inc10 setting**, the average accuracy $\bar{A}$ over five seeds is:
>
> - seed $17$: $86.16$
> - seed $204$: $86.67$
> - seed $1993$ (default in the submission): $86.01$
> - seed $7777$: $85.48$
> - seed $9527$: $86.15$
>
> This yields: $\bar{A} = 86.09\% \pm 0.43\%,$
> where the deviation is computed across the five random seeds.
>
> Two observations are important here:
> (i) the originally reported number ($86.01\%$ with seed $1993$) is very close to the multi-seed mean, indicating that our default seed is representative rather than a “lucky” run; and
> (ii) the between-seed variation (about $0.4$ percentage points) is much smaller than the accuracy gains that LACE achieves over the strongest baselines on CUB-200 in Table 2, so our conclusions about the effectiveness of LACE are not driven by randomness.
>
> ---
>
> **Reply to W2:**
> Thank you for raising the concern about ablation and interpretability analysis. We agree that it is important to disentangle the contribution of the Concept Attribution (CA) and Concept Verification (CV) components, and to provide qualitative evidence of interpretability.
>
> **(a) Ablation of CA and CV.**
> we have added an ablation study that isolates the effect of CA and CV. On **CUB-200 with ViT-B/16 in the B100 Inc10 setting**, we compare three variants:
>
> - **CA only**: LACE with CA but without CV (we use a fixed pruning ratio instead of data-driven verification);
> - **CV only**: LACE with CV but replacing CA with a naive magnitude-based ranking;
> - **CA + CV (full LACE)**: our proposed method using both modules.
>
> The average accuracy $\bar{A}$ is:
>
> | Dataset | Model    | Setting   | Components | $\bar{A}$ |
> |---------|----------|-----------|------------|-----------|
> | CUB-200 | ViT-B/16 | B100 Inc10 | CA        | 81.53     |
> | CUB-200 | ViT-B/16 | B100 Inc10 | CV        | 81.61     |
> | CUB-200 | ViT-B/16 | B100 Inc10 | CA + CV   | 82.64     |
>
> We observe that **both CA and CV individually improve performance**, but the **full LACE (CA + CV)** achieves the best result with $\bar{A} = 82.64\%$, which is about **+1 percentage point** higher than either CA-only or CV-only. This shows that CA and CV are complementary: CA provides principled concept importance scores (satisfying Sensitivity and Implementation Invariance), while CV uses a hat-matrix-based criterion to determine how aggressively to prune concepts on each task.
>
> We also add a qualitative analysis to illustrate which concepts are pruned versus retained by LACE. Figure 2 in the Appendix visualizes one CUB-200 class under **ViT-B/16**:
>
> - The **pruned concepts** (top panel) mostly describe background or contextual information, such as *“soar above the choppy, blue ocean waters nearby”*, *“pale pink”*, or *“sharp and slightly curved long, slender wings”*. These tend to capture scene context or generic motion descriptors that are not consistently discriminative for the target class across tasks.
> - The **retained concepts** (bottom panel) focus on stable, class-specific attributes of the bird itself, such as *“glossy black feathers that shimmer in the sunlight”*, *“eyes are small and framed by white feathers”*, and *“stands confidently on sturdy legs”*.
>
> The figure also shows representative images corresponding to these concepts on CUB-200. Qualitatively, CV tends to remove concepts tied to transient backgrounds or loosely related descriptions, while preserving concepts that describe consistent, discriminative morphology and appearance. This supports our claim that LACE yields a **more compact and semantically focused concept set** over time, improving interpretability while maintaining strong continual learning performance.

---

> ### Author Response · Authors · 2025-11-27
> **Reply to Question 1**
>
> **Reply to Q1:**
> Thank you for this question. We have added an explicit sensitivity study on (i) the pruning-rate bound and (ii) the per-task concept candidate cap. All experiments below are conducted on **CUB-200 with ViT-B/16** under the **B10 Inc10** and **B100 Inc10** settings.
>
> ---
>
> **(a) Varying the pruning-rate bound.**
> We vary the upper bound on the per-task pruning rate (i.e., the maximum fraction of concepts that can be removed on each task) while still letting CV choose the actual pruning rate within this bound. The average accuracy $\bar{A}$ is:
>
> - **B10 Inc10 and B100 Inc10**:
>
> | Pruning Rate | B10 Inc10 | B100 Inc10 |
> |---|---|---|
> | 0.05 | 86.14 | 82.41 |
> | 0.10 (default) | 86.01 | 82.64 |
> | 0.15 | 85.99 | 82.45 |
> | 0.20 | 85.97 | 82.36 |
> | 0.25 | 86.07 | 82.39 |
> | 0.30 | 85.78 | 82.33 |
> | 0.35 | 85.84 | 82.26 |
> | 0.40 | 85.57 | 82.26 |
> | 0.45 | 85.23 | 82.20 |
> | 0.50 | 85.35 | 82.19 |
> | 0.55 | 85.24 | 82.12 |
> | 0.60 | 84.75 | 82.07 |
> | 0.65 | 83.71 | 82.01 |
> | 0.70 | 83.98 | 81.95 |
> | 0.75 | 82.59 | 81.92 |
> | 0.80 | 81.59 | 81.85 |
> | 0.85 | 80.72 | 81.86 |
> | 0.90 | 79.28 | 80.97 |
>
>
> We observe that:
>
> - For both B10 Inc10 and B100 Inc10, **LACE is very stable when the pruning-rate bound is in a reasonable range**, roughly between $0.05$ and $0.30$. In this region, $\bar{A}$ varies within about $0.3$–$0.4$ percentage points.
> - Performance degrades only when we enforce **very aggressive bounds** (e.g., $\geq 0.5$), which effectively encourages over-pruning and removes too many concepts per task.
>
> Our default choice of a $0.10$ pruning-rate bound lies in this stable region and is close to the best-performing settings. These results indicate that LACE is **not overly sensitive** to the pruning-rate bound as long as we avoid extreme, unrealistic pruning.
>
> ---
>
> **(b) Varying the per-task concept candidate cap.**
> We also vary the per-task concept candidate cap, i.e., the maximum number of concepts that can be considered for each task before CV decides which ones to prune. The results on **B10 Inc10** are:
>
> | Concept Cap | $\bar{A}$ |
> |------------:|:---------:|
> | 1           |  75.86    |
> | 2           |  80.30    |
> | 3           |  82.67    |
> | 4           |  83.59    |
> | 5           |  84.37    |
> | 6           |  84.98    |
> | 7           |  85.46    |
> | 8           |  85.61    |
> | 9           |  86.18    |
> | 10 (default) | 86.01    |
>
> We see that:
>
> - When the cap is extremely small (e.g., $1$–$3$), $\bar{A}$ drops significantly because the model is forced to operate with too few concepts, limiting both representation power and interpretability.
> - Performance **rapidly improves and then saturates** once the cap reaches around $7$–$10$ concepts. In this regime, the differences in $\bar{A}$ are within about $1$ percentage point.
>
> Our default cap of $10$ concepts per task lies in this saturated region: increasing it further is unnecessary, while decreasing it too aggressively harms performance. Importantly, this also suggests that **LACE can work well with a relatively small number of concepts per task**, which is desirable from an interpretability perspective.
>
> ---
>
> **Summary.**
> Overall, these two ablations show that:
>
> 1. LACE is **robust** to reasonable choices of the pruning-rate bound and concept candidate cap;
> 2. Our default hyperparameters (pruning-rate bound $0.10$, concept cap $10$) lie in regions where performance has already plateaued; and
> 3. Even with modest concept caps, LACE maintains strong accuracy, supporting the claim that our method achieves a good trade-off between **compact concept sets** and **continual learning performance**.
>
> ---
>
> Thank you again for your thoughtful and constructive comments. We have carefully revised the paper to address all of your concerns. We hope these updates help clarify the strengths and validity of our method, and we warmly welcome any further questions or discussion.

---

### Author Response · Authors · 2025-11-27

We sincerely appreciate the reviewers’ detailed feedback and the numerous experimental suggestions. Because many of the requested analyses required substantial additional computation and careful redesign of experiments, we spent considerable time to ensure that every point was addressed as rigorously as possible.

We have now submitted a fully updated revision version of the manuscript, and the supplementary material includes a complete diff file that highlights all textual and mathematical changes. As shown in the diff, we have:

* corrected all inconsistent notations (e.g., prototypes, classifier weights, and temporary variables),
* expanded the explanations of **CA** and **CV**, including added intuition and clearer derivation steps,
* clarified ambiguous definitions and reorganized several subsections for better readability,
* added newly requested experiments, including ablations, fairness analyses, forgetting metrics, and compute-overhead measurements,


We hope that these revisions resolve the reviewers’ concerns and allow for a fair re-evaluation of our work. We would be very happy to discuss any remaining questions or clarifications.

---

### Meta-Review · Area_Chair_Zuvd · 2025-12-19

**Summary:**

All reviewers appreciated the motivation of the work and though the combination of concept bottleneck models and continual learning to be meaningfully conducted in the paper. In general, reviewers thought that the method has a solid grounding and mathematical motivation and that the intuition was very clear.

At the same time, all reviewers had several concerns, some of which were shared among reviewers. One of the bigger shared concerns revolved around the empirical validation of the work. Points of critique were lack of ablation studies, lack of analysis on efficiency and memory, lack of statistical significance in results, and potentially limited improvements as a consequence over other methods. Another concern shared across the reviews was inconsistency in notation, a confusing display of equations, and a generally dense and a hard to follow mathematical narrative that didn't motivate individual steps sufficiently.

As argued below, the AC believes that some of these parts were added in the rebuttal, but not all are reflected in the uploaded and thus not all major concerns of the reviewers could be clarified (even if there had been a discussion). Unfortunately, the AC thus recommends to reject the paper.

**Reviewer Concerns:**

Two primary concerns of the reviewers revolved around rigor, both with respect to the mathematical descriptions as well as the empirical results.

The rebuttal has largely focused on providing extensive additional results, ranging from select ablation studies to analysis on computational cost and comparison with other methods and metrics. Although the amount of added results is very large and thus somewhat hard to grasp as a change to the original version, the AC believes that many of the initial reviewer's concerns in this regard could have been addressed. However, some of the major concerns do not seem to be resolved in the uploaded pdf and the final message by the authors. The analysis of statistical significance could not yet be resolved fully, similar to the question on whether the improvements are "sufficient".

On the other hand, a big point of criticism in the reviews was the dense nature, inconsistent notation, hard to follow narrative with many steps missing. Looking at the rebuttal there are select comments on these aspects, but again carefully analyzing the revised pdf the main part of the paper is essentially identical. Some notation has indeed been corrected, but the AC believes many of the reviewers' points on clarity still stand. Importantly, the paper is no less "condensed" than it was before in this parts, with the main different being additional empirical results.

**Reviewer Scores:**

Reviewer scores were 2, 4, 4, 4 and thus all on the initially negative side of the spectrum. Unfortunately, due to the OpenReview incident, no discussion was possible between the authors and reviewers, but the authors went through a great extent to provide extensive rebuttals and uploaded a pdf revision.

Extrapolating reviewer scores as an AC on the basis of a potential discussion is naturally very challenging, so the AC has also read the paper.  After reading, the AC believes that some of the scores could have potentially been raised as a response to the rebuttal.
Specifically, the AC believes that the reviewers asking for empirical results could have raised their score by e.g. 2 points. In particular, it is likely that the reviewer who gave the score of 2 could have been satisfied with the rebuttal, given that the entire premise of the review was based on empirical results.

Unfortunately, it is not immediately clear to the AC that all scores would have shot up to a satisfactory level of all concerns having been resolved, even if a discussion had happened. On the one hand, many additional results were provided and some points could be clarified. On the other hand, the actually revised pdf doesn't look nearly as extensively changed as the rebuttals make it appear. Some additional appendix sections have been added, but the main paper remains largely the same (with the exception of fixed notation). The various reviewers concern regarding the understanding of the method and its narrative thus also make it likely that any increase in score would have been limited to minor changes at best. In particular, since one of the two main concerns seems to persist, the AC unfortunately does not believe the paper would have gotten accepted in a straightforward fashion.

Again, the AC understands that this is a complicated matter and therefore has read the main body parts of the paper. In the end, the AC thinks improvements have been suggested, but a more thorough revision is required for the paper to be truly improved. The AC understands that this may be disappointing to the authors and nevertheless appreciates the great amount of experimental effort put into the rebuttal. The AC is sure that with a more thorough revision of the structure and the addition of the new results, the paper will be in a good state for resubmission to a different venue.

---

### Decision · Program_Chairs · 2026-01-26

Reject